# Dynamic changes in the epigenomic landscape regulate human organogenesis and link to developmental disorders

Dave T. Gerrard [1,6], Andrew A. Berry[1,6], Rachel E. Jennings[1,2], Matthew J. Birket [1], Peyman Zarrineh[1], Myles G. Garstang[1], Sarah L. Withey[1], Patrick Short [3], Sandra Jiménez-Gancedo[4], Panos N. Firbas[4], Ian Donaldson [1], Andrew D. Sharrocks[1], Karen Piper Hanley [1,5], Matthew E. Hurles[3], José Luis Gomez-Skarmeta[4], Nicoletta Bobola[1] & Neil A. Hanley [1,2 ✉]

How the genome activates or silences transcriptional programmes governs organ formation. Little is known in human embryos undermining our ability to benchmark the fidelity of stem cell differentiation or cell programming, or interpret the pathogenicity of noncoding variation. Here, we study histone modifications across thirteen tissues during human organogenesis. We integrate the data with transcription to build an overview of how the human genome differentially regulates alternative organ fates including by repression. Promoters from nearly 20,000 genes partition into discrete states. Key developmental gene sets are actively repressed outside of the appropriate organ without obvious bivalency. Candidate enhancers, functional in zebrafish, allow imputation of tissue-specific and shared patterns of transcription factor binding. Overlaying more than 700 noncoding mutations from patients with developmental disorders allows correlation to unanticipated target genes. Taken together, the data provide a comprehensive genomic framework for investigating normal and abnormal human development.

[1] Faculty of Biology, Medicine & Health, Manchester Academic Health Sciences Centre, University of Manchester, Oxford Road, Manchester M13 9PT, UK. [2] Endocrinology Department, Manchester University NHS Foundation Trust, Grafton Street, Manchester M13 9WU, UK. [3] Wellcome Sanger Institute, Wellcome Genome Campus, Hinxton, UK. [4] Centro Andaluz de Biología del Desarrollo (CABD), Consejo Superior de Investigacionnes Cientificas/ Universidad Pablo de Olavide/Junta de Analucía, Sevilla, Spain. [5] Wellcome Centre for Cell-Matrix Research, University of Manchester, Oxford Road, Manchester M13 9PT, UK. [6] These authors contributed equally: Dave T. Gerrard, Andrew A. Berry. ✉email: neil.hanley@manchester.ac.uk

Organogenesis is the key phase when the body's tissues and organs are first assembled from rudimentary progenitor cells. In human embryos, this is the critical period during weeks 5–8 of gestation when disruption can lead to major developmental disorders. While ~35% of developmental disorders are explained by damaging genetic variation within the exons of protein-coding genes[1], de novo mutation (DNM) in the non-coding genome has been associated increasingly with major developmental disorders[2]. The non-coding genome also harbours over 80% of single-nucleotide polymorphisms (SNPs) implicated in genome-wide association studies (GWAS) for developmental disorders, or in GWAS of later onset disease, such as schizophrenia and type 2 diabetes, where contribution is predicted from early development[3]. These genetic alterations are presumed to lie in enhancers for developmental genes or in other regulatory elements, such as promoters for non-coding RNAs that may only be active in the relevant tissue at the appropriate stage of organogenesis. Aside from rare examples[4], this has remained unproven because of lack of data in human embryos. While regulatory data are available in other species at comparable stages[5], extrapolation is of limited value because the precise genomic locations of enhancers are poorly conserved[6,7] even allowing for enriched sequence conservation around developmental genes[8,9]. Sequence conservation alone is also uninformative for when and in what tissue a putative enhancer might function. Comprehensive regulatory information is available from later fetal development via initiatives such as NIH Roadmap[10], but these later stages largely reflect terminally differentiated, albeit immature cells rather than progenitors responsible for organ formation. In contrast, a small number of studies on a handful of isolated tissues, such as limb bud[11], craniofacial processes[12], pancreas[13], or brain[14], have demarcated regulatory elements directly during human organogenesis. However, most organs remain unexplored. Moreover, nothing is known about patterns of regulation, including both activation and repression, deployed across tissues, which is an important factor because tissues are often co-affected in developmental disorders. To address these gaps in our knowledge, we set out to build maps of genome regulation integrated with transcription during human organogenesis at comprehensiveness currently unattainable from single-cell analysis.

## Results

**Assembly of datasets across 13 tissues**. Thirteen different types of organs and tissues were contributed by 61 human post-implantation embryos, microdissected and subjected to chromatin immunoprecipitation followed by deep sequencing (ChIPseq) for three histone modifications (Fig. 1a): H3K4me3, enriched at promoters of transcribed genes; H3K27ac, at active enhancers and some promoters; and H3K27me3 delineating regions of the genome under active repression by Polycomb. Tiny tissue size and the scarcity of human embryonic tissue required some pooling and precluded study of additional modifications. Biological replicates were undertaken for all but two tissue sites (details are in Supplementary Data 1 and 2). These all included both male and female tissue. Tissues and stages were matched to polyadenylated RNAseq datasets acquired at sufficient read depth to identify over 6000 loci with previously unannotated transcription (Supplementary Data 3)[15]. Overlaying the data revealed characteristic tissue-specific patterns of promoter and putative enhancer activity, and unannotated human embryonic transcripts. This was particularly noticeable surrounding genes encoding key developmental transcription factors (TFs), such as the example shown for *NKX2-5* in the heart (Fig. 1b). Tissues lacking expression of the TF gene tended to carry active H3K27me3 modification rather than simply lack marking.

Putative tissue-specific enhancer marks were characteristically distributed over several hundred kilobases (heart-specific peak (red) over 200 kb from *NKX2-5* to the far right of Fig. 1b). These isolated H3K27ac marks were often unpredicted by publicly available data from cell lines or terminally differentiated lineages and did not necessarily show sequence conservation across vertebrates (mean per-base phyloP score 0.175; range −1.42 to +6.94 for $n = 51,559$ regions)[16]. Unexpected H3K4me3 and H3K27ac peaks that failed to map to the transcriptional start sites (TSSs) of annotated genes mapped to the TSS of previously unidentified human embryo-enriched transcripts, such as the example shown in Fig. 1b for the bidirectional *HE-TUCP-C5T408* and *HE-LINC-C5T409* (for the complete catalogue see Supplementary File 1H in ref. [15]). Recognising the importance of features surrounding key developmental genes such as *NKX2*-5, all the individual data are available to browse as tracks on the UCSC Genome Browser. These preliminary observations encouraged us to undertake full integration of the different datasets to enable a series of genome-scale analyses.

**Major promoter states partition without obvious bivalency**. By analysis based on a Hidden Markov Model, the genome partitioned into different chromatin states very similarly across tissues[17]. While three histone marks allowed for eight different segmentations, aggregation into fewer states was possible (Fig. 1c). On average across tissues, 3.3% of the genome was active promoter (States 1 & 2; H3K4me3 +/− H3K27ac) or putative enhancer (State 3; H3K27ac) (range 1.7–6.1%; Fig. 1c & Supplementary Fig. 1). In all, 6.7% was variably marked as actively repressed (States 6 & 7; range 3.3–13.0; H3K27me3), while on average 89.8% of the genome was effectively unmarked (States 4 & 5; range 81.7–94.0). Approximately 0.2% seemingly had both H3K4me3 and H3K27me3 marks (State 8; range 0.16–0.33). This latter state has been considered bivalent and characteristic of poised genes whose imminent expression then initiates cell differentiation pathways[18–20]. Ascribing bivalency has been reliant on setting an arbitrary threshold for a binary decision of whether a site is marked or not. This risks the impression of equivalence when in fact one or other mark might be far more prominent. Moreover, apparent bivalency could simply reflect mixed marks due to heterogeneity in a cell population (the minor co-detection of H3K27ac in State 8 would not be expected in the presence of H3K27me3). Taken together, this reliance on an arbitrary threshold is suboptimal. Therefore, we used ngsplot to cluster promoter profiles for each histone mark integrated with transcription over 3 kb either side of 19,791 distinct protein-coding TSS in each tissue[21] (Fig. 2; Supplementary Figs. 2–4). We started under default settings for three histone modifications and RNA-seq, which generated the five most prominent clusters. Broader H3K4me3 and H3K27ac signals at the TSS correlated with higher levels of transcription (Fig. 2a–b; we termed this promoter state broad expressed versus narrow or bidirectional expressed). In total, 25–30% of genes across tissues were unmarked and lacked appreciable transcription (inactive). These promoters typically lacked CpG islands (<20% compared with 67.7% of the 19,791 genes). Conversely, 90–95% of TSS regions marked with H3K27me3 featured CpG islands with an over-representation of TFs; 31.2% of TFs ($n = 1659$) were actively repressed in at least one tissue compared with 20.0% of non-TF genes (odds ratio 1.82, confidence interval 1.63–2.04; P-value < 2.2e-16). H3K27me3 detection at the TSS was ~50% greater for genes encoding TFs (Supplementary Fig. 5). The categorisation for each of the 19,791 genes in all tissues is listed in Supplementary Data 4. When prioritised under default settings, patterns across tissues were strikingly similar with only minor variation in

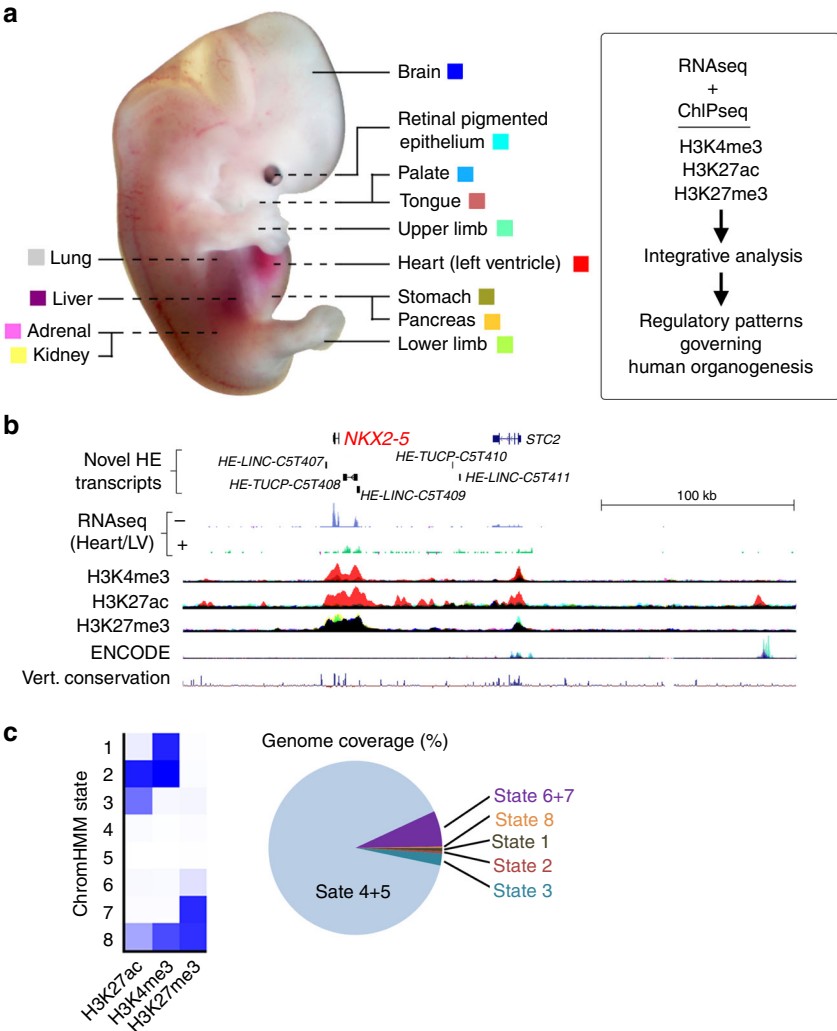

**Fig. 1 Epigenomic landscape across 13 human embryonic tissues. a** Thirteen different human embryonic sites were sampled for RNAseq[15] and ChIPseq, as described in the "Methods" and in Supplementary Data 1 and 2. The same colour coding for each tissue is applied throughout the paper in overlaid ChIPseq tracks. The heart (left ventricle) dataset is summarised as Heart/LV from hereon. **b** 300 kb locus around the *NKX2-5* gene, the most discriminatory TF gene for human embryonic heart[15]. The locus contains five unannotated human embryonic (*HE*) transcripts enriched in heart [three *LINC* RNAs and two transcripts of uncertain coding potential (*TUCP*)]. Heart/LV-specific (red) H3K4me3 and H3K27ac marks were detected at the *NKX2-5* TSS and adjacent transcripts (*HE-TUCP-C5T408* and *HE-LINC-C5T409*). Embryonic heart-specific H3K27ac marks were visible up to 200 kb away (e.g., at the extreme right of panel). H3K27me3 marked the region from *NKX2-5* to *HE-LINC-C5T409* in all non-heart tissues (the track appears black from the superimposition of all the different colours other than red). ENCODE data are from seven cell lines[26]. **c** Genome coverage by ChromHMM for the different histone modifications was similar across all tissues (Supplementary Fig. 1) with an average 89.8% of the genome unmarked (range: 81.7–94.0; States 4 & 5), and 3.3% consistent with being an active promoter and/or enhancer (range: 1.7–6.1; States 1–3).

bidirectional transcription in RPE (Supplementary Fig. 2) and a technical factor limiting the detection of expression at the TSS for particularly long transcripts in the liver, lung and brain (Supplementary Figs. 3 and 4). Consistently, neat partitioning would not have been possible if the data were overly confounded by cellular heterogeneity. While a major bivalent chromatin state at gene promoters was not detected under these parameters, it was noticed that H3K4me3 levels were slightly higher in the active repression group than for inactive genes (Fig. 2b, H3K4me3, red and grey lines, respectively). H3K27ac signal for active repression and inactive virtually overlapped, very close to the background signal. Therefore, we extended the ngsplot parameters to allow more subcategorization in the search for bivalency. Explicitly, we wanted to see if we could split the H3K27me3 signal into a subset of characteristic bivalent genes with clear cut H3K4me3. Seven

(kidney), eleven (liver), or ten categories (all other replicated tissues) split the H3K27me3 signal. For each tissue, the smaller H3K27me3 sub-category now clustered with weak detection of H3K4me3 (and some low-level transcription). We assessed enrichment of these genes in each tissue for putative bivalent characteristics, such as imprinting or TFs characteristic of the particular tissue fate. However, in all instances the same generic categories, such as extracellular matrix organisation, receptor tyrosine kinases and a variety of neural terms emerged. Supplementary Figs. 6–11 show examples for the adrenal, kidney and pancreas. These categories and the genes underlying them are well-recognised to reflect mesenchyme and neural development as has been detailed for the pancreas[22] with characteristic WNT and FGF family members[23]. In summary, promoter regions partitioned neatly into major chromatin states allied to

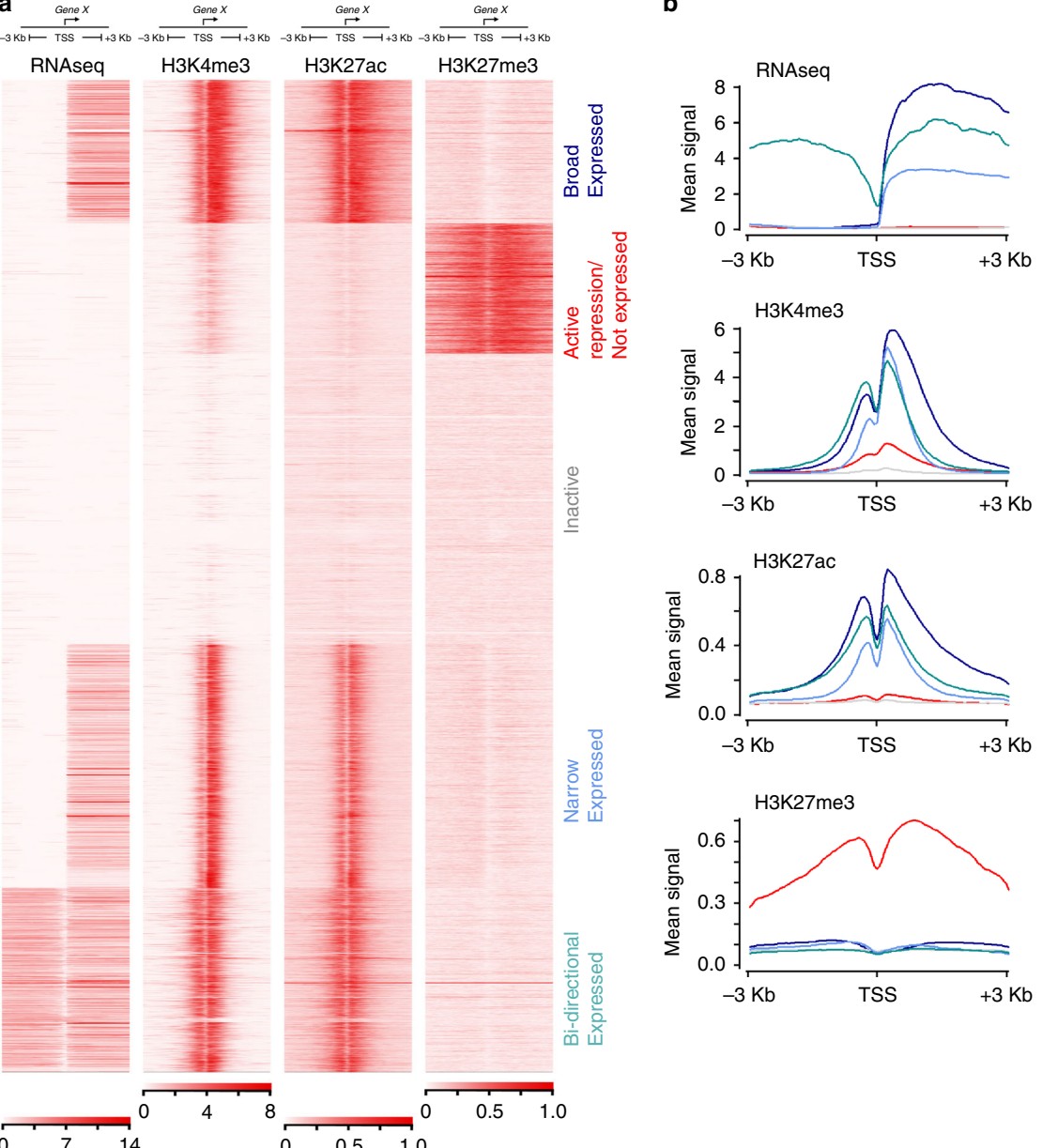

**Fig. 2 Classification of major promoter states. a** Clustered heatmaps surrounding the transcriptional start sites (TSS +/− 3 kb) of 19,791 annotated genes. The example shown is for adrenal. One replicate is shown for each data type for simplicity. Replicates across all tissues were near identical. Two minor variations on this pattern were detected in RPE (Supplementary Fig. 2) and the liver, lung and brain (Supplementary Fig. 3). **b** Mean signal levels for the genes clustered in (**a**). Traces are coloured according to the text colour in (**a**). Broad expressed genes show approximately double the level of transcription and twice the width of H3K4me3 and H3K27ac marks compared with narrow expressed genes.

transcription. Sets of bivalent genes were not observed before gene sets indicative of cell-types common across all tissues limited the extent of subcategorization.

**Mapping major promoter states discovers disallowed gene sets.** Our classification allowed us to ask how the major promoter states changed across different tissues. Tracking all states in all tissues was complex to visualise (Supplementary Fig. 12). Unifying broad, narrow and bidirectional expressed into a single category (expressed) displayed how the majority of genes remained unaltered across tissues (Fig. 3a). In contrast, 29% of genes had a variable promoter state. Within this subset, we predicted that genes responsible for a specific organ's assembly, such

as those encoding developmental TFs, would need to be actively excluded or disallowed at inappropriate sites (as seen in Fig. 1b for *NKX2-5*). We tested this in the replicated datasets by comparing genes transcribed uniquely in one tissue for either inactivity (no mark) or active repression elsewhere (H3K27me3; disallowed). Gene ontology (GO) analysis of the uniquely expressed/disallowed elsewhere gene sets identified the appropriate developmental programme in all instances (as shown for the heart in Fig. 3b; e.g., heart development). These gene sets are listed for each tissue in Supplementary Data 5a. In contrast, tissue-specific transcription initiated from genes that were simply inactive in other organs tended to highlight differentiated cell function (Fig. 3b; e.g., sarcomere organisation). These observations highlight the preferential use of H3K27me3 at the

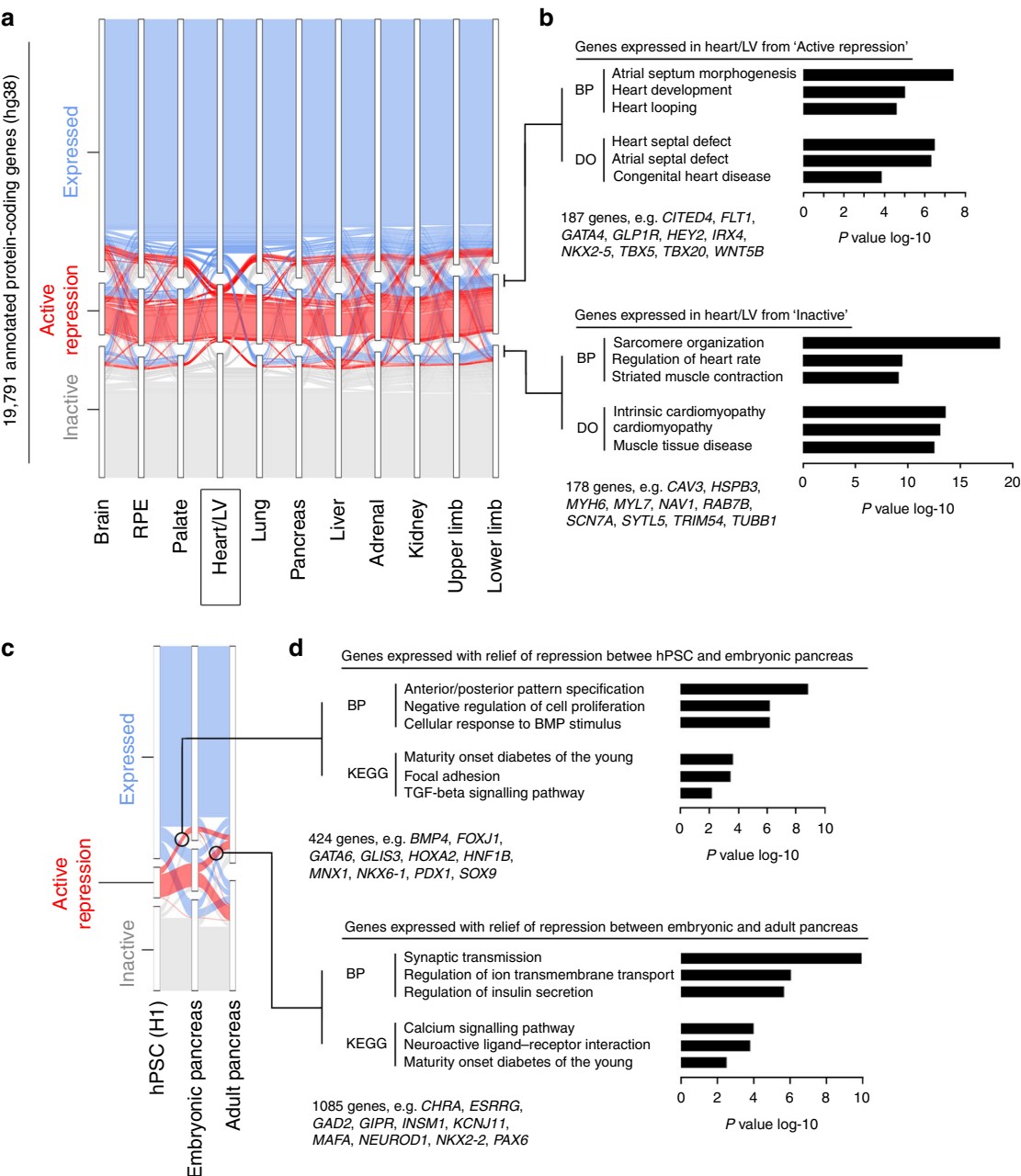

**Fig. 3 Integration of promoter states across tissues and over time. a** Alluvial plot showing promoter state for 19,791 annotated genes across all tissues with replicated datasets. To aid visualisation, all the different transcribed states are amalgamated into a single expressed category (the alluvial plot for all individual states is shown in Supplementary Fig. 6). The example shown is centred on the promoter state in the Heart/LV dataset. Those genes with an expressed promoter state in the heart and either active repression or inactive elsewhere are indicated to the right of the panel and subject to gene enrichment analyses in (**b**). **b** Gene enrichment analysis of genes with an expressed promoter state in the heart and either active repression or inactive in all remaining tissues. All remaining genes were used as background. Examples of the genes underlying the biological process (BP) or disease ontology (DO) terms and their total number are listed beneath the bar charts. **c** Alluvial plot showing the variance in promoter state between H1 human pluripotent stem cells (hPSCs), the embryonic pancreas (prior to endocrine differentiation[24]) and the adult pancreas. Circles capture those genes that shift from active repression to expressed at the stage of either embryonic or adult pancreas. **d** Gene enrichment analyses of encircled genes from (**c**). Examples of the genes underlying the BP and KEGG terms and their total number are listed beneath the bar charts. All remaining genes were used as background. While maturity onset diabetes of the young emerged in both analyses, the underlying genes were different reflecting developmental roles prior to or after pancreatic endocrine differentiation[24].

promoters of genes controlling cell fate decisions, but not differentiated function. Moreover, the fact that these cell fate genes did not emerge for any tissue in the sub-categorisation of H3K27me3 signal with low-level H3K4me3 (Supplementary Figs. 6–11) further supports lack of bivalency at their gene promoters.

To scrutinise regulatory changes temporally, we focussed on pancreas and included datasets from human pluripotent stem cells (hPSCs) and adult tissue. Different sets of repressed genes lost their H3K27me3 mark to become expressed as cells transitioned from pluripotency to embryonic pancreatic progenitors or from pancreatic progenitors to mature pancreas (Fig. 3c;

genes are listed in Supplementary Data 5b). Surprisingly, the same KEGG term relating to monogenic diabetes emerged in both instances (Fig. 3d). However, the genes underlying the first transition related to early function in pancreatic organogenesis, hypoplasia, or aplasia (e.g., *GATA6*, *SOX9* and *PDX1*); while the genes in the second transition specifically related to post-embryonic pancreatic islet cell differentiation and beta-cell function (e.g., *INSM1*, *MAFA* and *NKX2-2*)[24,25].

**Human regulatory sequences function in zebrafish embryos.** Having recognised the disallowed status of developmental TFs in inappropriate tissues, we wanted to test whether our putative intergenic human embryo-enriched enhancers were capable of driving appropriate reporter gene expression at the correct locations in developing zebrafish. We identified H3K27ac marks that were enriched in the human embryo compared with 161 ENCODE or NIH Roadmap datasets[10,26] and not detected in the FANTOM5 project[27]. We developed an algorithm to test for embryonic tissue specificity and filtered for sequence conservation (not necessarily in zebrafish; see "Methods"). We manually inspected the remainder for proximity (<1 mb) to genes encoding TFs and, in particular, to increase clinical relevance, to those associated with major developmental disorders. We ensured no H3K4me3 or polyadenylated transcription in the immediate vicinity (i.e., an unannotated promoter). We tested 10 such enhancers out of 44 within 1 mb of *TBX15*, *HEY2*, *ALX1*, *IRX4*, *PITX2*, *HOXD13*, *NKX2-5*, *WT1*, *SOX11* and *SOX9* for their ability to direct appropriate GFP expression in stable lines of transgenic zebrafish (Supplementary Data 6). Two (h-003-kid near *WT1* and h-022-mix near *SOX11*) failed to generate any GFP in any location. The remaining eight all yielded GFP at the predicted site in zebrafish embryos (Fig. 4; Supplementary Data 6), despite only one of the putative enhancer sequences being conserved in zebrafish (Fig. 4a, h-027-lim near *TBX15*). Taken together, these data imply that our H3K27ac detection marks previously unannotated human enhancers, which function over considerable evolutionally distance, despite the absence of detectable sequence conservation. This is most likely associated with overall conservation of TF activity between homologous organs across species[28].

**Patterns of H3K27ac modification are unique across tissues.** We wanted to explore the link between these regulatory elements and surrounding gene expression at genome-wide scale. Assured that ChIPseq marks were reproducible within biological replicates without batch effect (Supplementary Fig. 13), we parsed the genome into 3,087,584 non-overlapping 1 kb bins. Reads within each bin were counted for each mark. Phi correlation between biological replicates indicated this approach to peak calling was very similar to using MACS (Supplementary Fig. 14). Counts were downsampled and averaged within tissues and correlated with the corresponding RNAseq data over 1 mb in either direction (i.e., a 2 mb window). On average, this window included 44 annotated genes (range: 0–247 genes). For those H3K27ac marks which functioned in zebrafish, the strongest correlation was with the appropriate TF gene, for instance *TBX15* over ~500 kb in limb (Fig. 4a). Moreover, within the 2 mb window, different H3K27ac marks could be correlated to the same gene, potentially allowing previously unknown enhancers to be grouped, for instance in the adrenal around the adrenal hypoplasia gene, *NR0B1*, located on the X chromosome (Supplementary Fig. 15).

Parsing the ChIPseq data into bins allowed integration of information across tissues, which is challenging when based on empirical modelling by MACS. Placing raw read counts per bin in rank order produced near identical elbow plots for all marks in all

tissues. This avoided arbitrary assignment and allowed the point of maximum flexure to be used quantitatively for calling marks in a binary yes/no fashion (Fig. 5a). The simplified calling facilitated exploration of regulatory patterns across tissues. Requiring a bin to be marked in any two or more samples identified 48,570 different H3K27ac patterns genome wide. The top 40 are shown in Fig. 5b. While tissue specificity for the heart/left ventricle was the most common pattern, all replicated organs ranked within the top 0.6% of patterns. Nine out of the 11 replicated tissues ranked in the top 0.2% (Supplementary Data 7). These data indicate high reproducibility and consistency across tissue replicates. H3K27ac showed far more tissue-selective patterns than H3K4me3 or H3K27me3 (Fig. 4b; Supplementary Figs. 16 and 17). Motif analysis on the tissue-specific H3K27ac regions allowed imputation of master TFs for individual tissues, such as NR5A1 in 54.5% of adrenal-specific bins ($n = 18,411$) compared to 25% of the remaining 141,706 bins (Fig. 5c). Mutation of *NR5A1* causes adrenal agenesis in human and mouse (OMIM 184757). MEF, TBX and bHLH family members emerged in the heart-specific bins (Fig. 5c); all are associated with congenital heart disease[29]. This emergence of TFs with known, critical tissue-specific functions underscores the validity of parsing the ChIPseq data into bins as well as the consistency of data between tissue replicates.

Having integrated our data, we could also uncover regulatory regions that were shared precisely across two or more tissues relevant to developmental disorders which manifest in multiple organs. Enrichment for composite PITX1/bHLH motifs was found in the limb and palate (Fig. 5c). GATA-binding motifs were enriched in the heart and pancreas. Shared patterns could be explicitly instructed by requiring detection in four or more samples (Supplementary Fig. 18). We hypothesised that patterns shared across many tissues ought to contain elements regulating generic developmental functions. Scrutinising bins marked in over half of all H3K27ac samples ($n = 30,226$ bins versus remaining background of 80,352 bins) identified enrichment for the ETS motif. ETS transcription factors are involved in cell cycle control and proliferation[30].

**Overlaying the epigenomes with developmental disorders.** Non-coding mutations in promoters or enhancers have been linked increasingly to major developmental disorders[4,31]. Previously, as part of the Deciphering Developmental Disorders (DDD) study, we studied 7930 individuals and their parents[32]. In all, 87% of patients had neurodevelopmental disorders. 10% had congenital heart defects. 68% of patients lacked disease-associated DNMs within exomes (exome-negative) pointing to the likely importance of the non-coding genome[2]. We sequenced 6139 non-coding regions (4.2 mb) selected as ultra-conserved regions (UCRs: $n = 4307$), experimentally validated enhancers (EVEs: $n = 595$) or as putative heart enhancers (PHE: $n = 1237$) and found 739 non-coding DNMs[2]. In total, 78% of the 6139 regions were marked by H3K27ac or H3K4me3 in our embryonic tissues, with a higher percentage overlap for the EVEs (87%) and near-perfect overlap for the PHEs (99%) (Fig. 6a). An additional 9% were marked by H3K27me3, suggesting non-coding regulation in a currently unsampled tissue. The distribution of DNMs was very similar (Fig. 6b). Nearly half of the regions containing DNMs were marked by H3K27ac and/or H3K4me3 that was replicated in at least one tissue. Most commonly, this included the heart or brain, in keeping with the predominance of neurodevelopmental and cardiac phenotypes in the DDD cohort and the PHEs selected for sequencing (Fig. 6c). In total, 75% of the PHEs with DNMs mapped to replicated H3K27ac and/or H3K4me3 in our heart dataset. This rose to 100% if the need for replication was

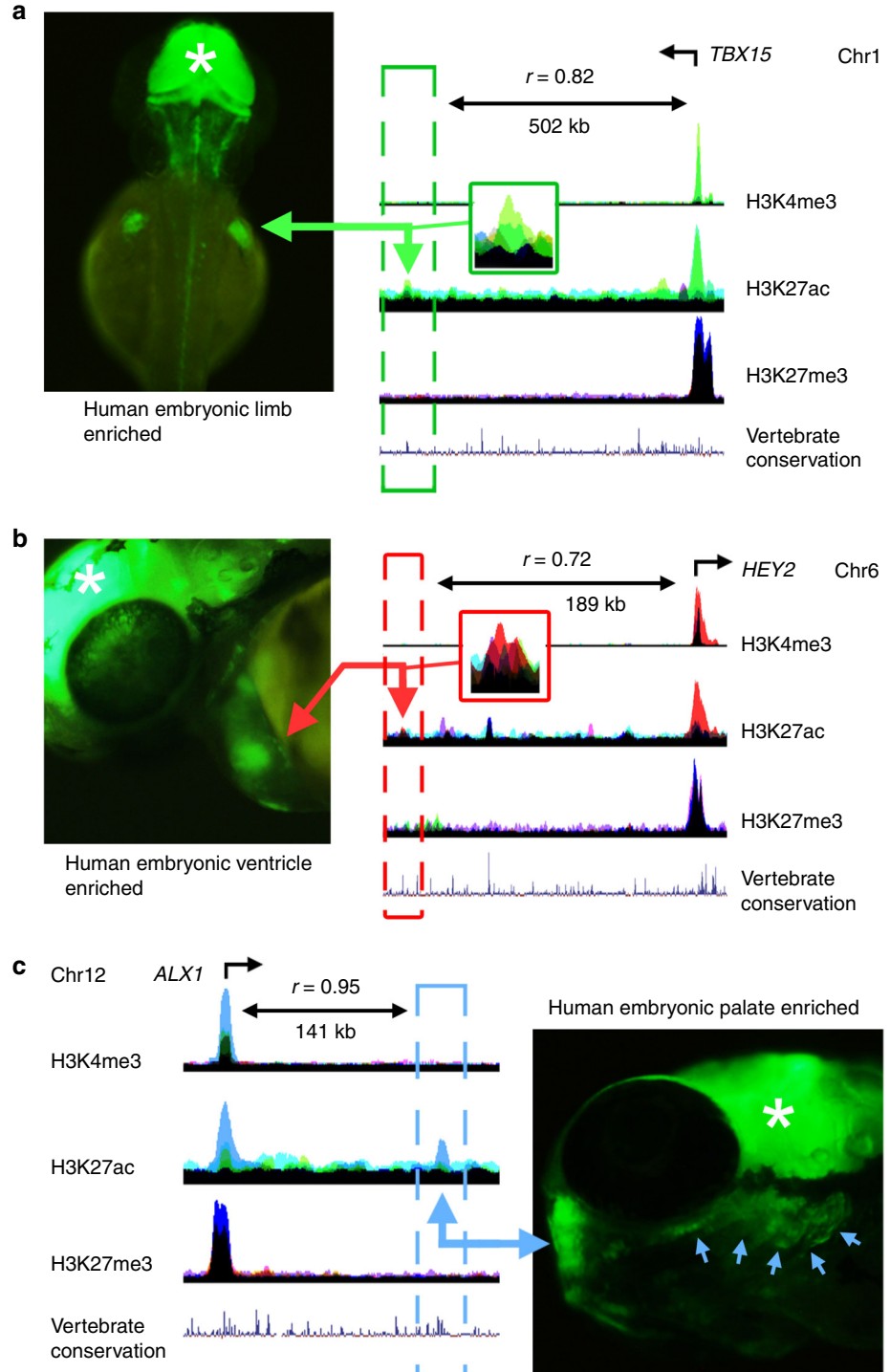

**Fig. 4 Transgenic analysis of H3K27ac regions from human embryonic tissues.** H3K27ac-marked regions were tested in multiple lines of stable transgenic zebrafish (details in Supplementary Data 4; same colour coding of tracks as in Fig. 1). **a** 231 bp limb enhancer, 502 kb downstream of *TBX15*, with the corresponding green fluorescent protein (GFP) detection in fin bud at 48 h post fertilisation (hpf). **b** About 355-bp heart/LV enhancer, 189 kb upstream of *HEY2*, with the corresponding ventricular GFP detection at 48 hpf. **c** 1.5 kb palate enhancer, 141 kb downstream of *ALX1*, with GFP in the developing trabecula and mandible (blue arrows) at 48 hpf. Correlations between the enhancer and transcription of the TF gene are shown for each example. Note the H3K27me3 marks over the gene in each instance in other tissues. *, midbrain GFP expression from the integral enhancer in the reporter vector used as a positive control for transgenesis.

removed. We did not observe enrichment for DNMs in patients with heart, eye or limb phenotypes in elements marked by H3K27ac. Similarly, organ-level association with clinical phenotype was not possible. In both circumstances, the power of testing was markedly curtailed by limitations in patient numbers

(Fig. 6d). Enrichment for DNMs in elements marked by H3K27ac was detected with neurodevelopmental disorders (87% of the DDD cohort; 1.45-fold, 95% confidence interval 1.09–1.90; $P = 0.0056$) (Fig. 6d). This was similar to our previous report using NIH Roadmap H3K27ac and/or DNaseI hypersensitivity data

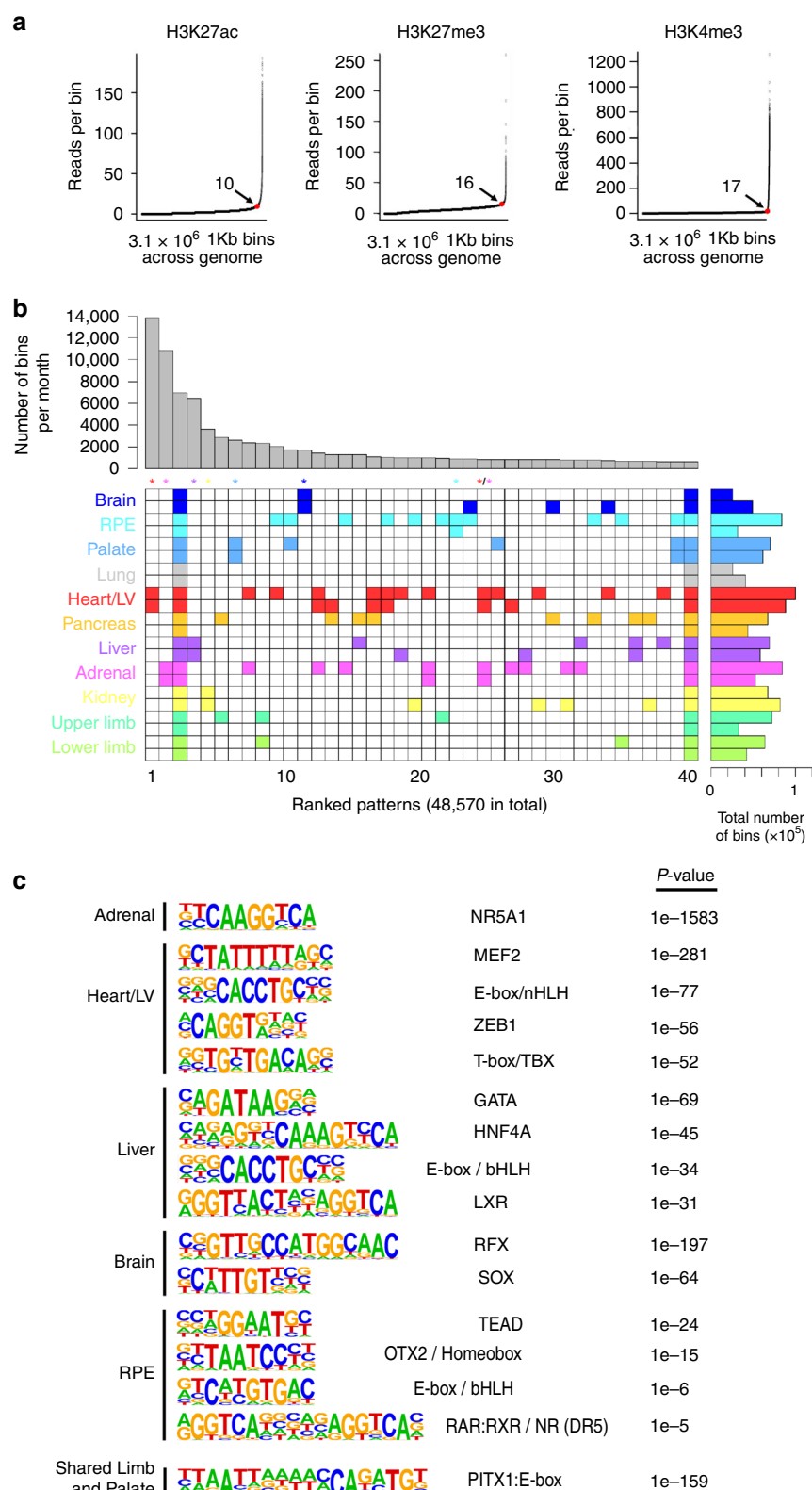

derived from second trimester fetal brain[2]. Our results support a role for non-coding mutations in severe neurodevelopmental disorders.

Prioritising individual DNMs for potential pathogenicity and how they might disrupt surrounding gene function is very challenging. We anticipate an atlas of human embryonic regulatory marks comprising different tissues will help stratify disease-relevant non-coding regions; and that correlation to surrounding transcription offers a preliminary means for prioritising putative target gene(s). We have made all correlations freely available as tracks on the UCSC Genome Browser and present two examples here as preliminary exemplars. A DNM linked to neurodevelopmental disorders in a UCR on chromosome 16 is in the middle of the annotated testicular LINC RNA,

**Fig. 5 Patterns of enhancer activity and transcription factor binding across tissues. a** Elbow plots for each histone modification following allocation of the genome into 3.1 million consecutive bins of 1 kb. The example shown is for adrenal providing the number of reads per bin at the point of maximum gradient change (the elbow point, red dot) and a quantitative measure of whether a bin was marked or not (e.g., >10 or <10, respectively, for H3K27ac). Converting marks into a binary yes/no call at any point in the genome facilitated the data integration across the different tissues. While the number of reads per bin at the elbow point was different for each mark across the tissues, the shape of the curve remained the same. **b** Euler grid for bins marked by H3K27ac (defined by elbow plots) in replicated tissues (i.e., two rows/replicates per tissue). Total number of marked bins per individual dataset is shown to the right. The example in (**b**) required a bin to be called in any two or more samples and is ordered by decreasing bin count per pattern (bar chart above the grid). A total of 48,570 different patterns were identified, of which the top 40 are shown. Tissue specificity for all sites emerged in the top 265 (0.5%) patterns; colour-coded asterisks above columns). For example, nearly 14,000 bins marked only in the two Heart/LV H3K27ac datasets ranked first as the most frequent pattern. The seventh most frequent pattern in ~3000 bins was palate-specific. Tissue-specific patterns were far less apparent at promoters (H3K4me3, $n = 18,432$; Supplementary Fig. 10) or for H3K27me3 ($n = 26,339$; Supplementary Fig. 11). While patterns across multiple tissues were permitted by stipulating marks in ≥2 samples (e.g., heart and adrenal in column 24), they could be enforced by stipulating marks in at least four samples (Supplementary Fig. 12). **c** Enrichment of known TF-binding motifs in the tissue-specific patterns of H3K27ac identified in (**b**). Five individual tissues are shown as examples alongside analysis of the shared regulatory pattern identified for the limb and palate identifying marked enrichment of a compound PITX1:E-box motif. Motif-enrichment was conducted using a one-sided Binomial test implemented in findMotifsGenome.pl of the HOMER package.

*LINC01572* (Fig. 7; chr16:72,427,838). Our data illustrate that the DNM is also located at the TSS of *HE-OT-AC004158.3*, expressed at 19.5-fold higher levels in human embryonic brain than any other tissue (mean read count of quantile normalised transcripts in the brain, 1317.2; mean in other tissues, 32.2), in a 4 kb region of brain-specific H3K27ac (and to a lesser extent, H3K4me3). Across 18 protein-coding genes in the region, the H3K27ac signal was most highly correlated to expression of *ZNF821* ($r = 0.92$) 550 kb away and anti-correlated to the adjacent gene, *ATXN1L* ($r = -0.65$; Fig. 7). As a second example, a cardiac-specific enhancer in the final intron of *ATXN1* on chromosome 6 contained a DNM from the DDD cohort associated with congenital heart disease (Fig. 8a). Across a 2.4 Mb region surrounding this putative enhancer, expression of the gene, *RBM24*, located ~1 mb away, was markedly enriched in heart compared to other organs (Fig. 8b) and the gene most correlated to the enhancer ($r = 0.88$). *RBM24* is required for cardiac development with knockout mice dying at E12.5-14.5 with ventricular septal defects and compromised cardiac muscle assembly[33]. We used CRISPR-Cas9 to delete the enhancer in our established *NKX2-5*-GFP reporter hPSC model[34,35] (Fig. 8c). Cells were viable, but loss of the enhancer markedly curtailed the generation of *NKX2-5*-positive cardiomyocyte progenitors (Fig. 8d–e). This was associated with markedly reduced expression of *RBM24*, but not of other genes in the locus (Fig. 8f).

Taken together, we have assembled datasets of regulatory activity linked to transcription for a range of human embryonic tissues during the period of organogenesis. All data are available to browse as tracks on the UCSC Genome Browser ready for overlay with genetic variants identified by clinical sequencing and GWAS.

## Discussion

Previous studies of enhancer usage in human embryos have tended to focus on individual tissues inferring, amongst other findings, aspects of genome regulation responsible for human-specific attributes[11–14]. Here, we incorporated epigenomic data with transcription across 13 sites during human organogenesis to build tissue-by-tissue maps of enhancers and promoters linked to gene expression. While similar to prior work in mouse[5] and building on our previous transcriptomic atlas[15], the integrated approach here offers opportunities to understand how human organ formation is regulated in health and disease.

It is important to draw out certain features of the study, including limitations. The datasets comprised pooled tissue samples across several embryos including both sexes. This removed the opportunity for analysis of male or female tissue in isolation; however, it mitigated against the risk of misinterpreting organ-level differences that might be due to sex, for instance related to H3K27me3 and X chromosome inactivation. Moreover, the aggregated data across many embryos reduced the risk of other misleading conclusions due to one-off biological or technical factors, such as deterioration during transport. Our tissue collection and molecular analyses occur on a single campus with immediate processing. For instance, in quality control, we have found overnight delay in preservation media can lead to a 16-fold decline in the levels of some key developmental TFs, despite tissue remaining viable and suitable for immunohistochemistry and subsequent tissue culture. It is also obvious that the bulk analyses contain aspects of tissue heterogeneity, as we reached in all tissues with our deeper analyses of promoter state (Supplementary Figs. 6–11). However, the emergent features of our data such as the gene ontology for major promoter states (Fig. 3b, d) and the binding motifs within tissue-specific enhancers (Fig. 5c) indicate the majority contribution from organ-specific progenitor cells. Our past analysis based on variance also minimises any contribution from cell-types common to all tissues[15]. While single-cell technology allows deconvolution of organ-level heterogeneity, at present the techniques do not permit the extent of analysis for histone modification or transcription as we have undertaken here. We envisage our integrated atlas will provide a valuable complement to current and future single-cell analyses of human organogenesis.

Assessing the non-coding genome is very challenging: millions of rare variants are returned from whole-genome sequencing (WGS) in each individual, while only one might be pathogenic[36]. Although we provided proof-of-principle here for a cardiac enhancer, functional analysis, even of a handful of variants, is clinically impractical. For non-coding mutations to affect organogenesis (either in developmental disorders or in later life disease such as type 2 diabetes where there is an embryonic contribution), it is logical that mutations are located in regulatory regions of the genome that are active in post-implantation human embryos. As evident from Fig. 1c, our identification of this landscape offers a timely pipeline for stratifying 98.5% of the genome down to 3% on average per tissue (States 1–3). At present, the associations from our dataset to non-coding mutations from patients is hampered by statistical power in the clinical cohorts, potentially allied to aspects of ascertainment bias, for instance towards neurodevelopmental disorders in DDD[32]. As cost continues to decline, WGS will become an increasingly important tool in main stream clinical investigation, opening up potential genetic diagnoses in the 98.5% of the human genome that lies outside of coding sequences. At this point, we anticipate that our datasets will help to stratify disease-associated variants. Enrichment of tissue-specific TF binding in these enhancers and

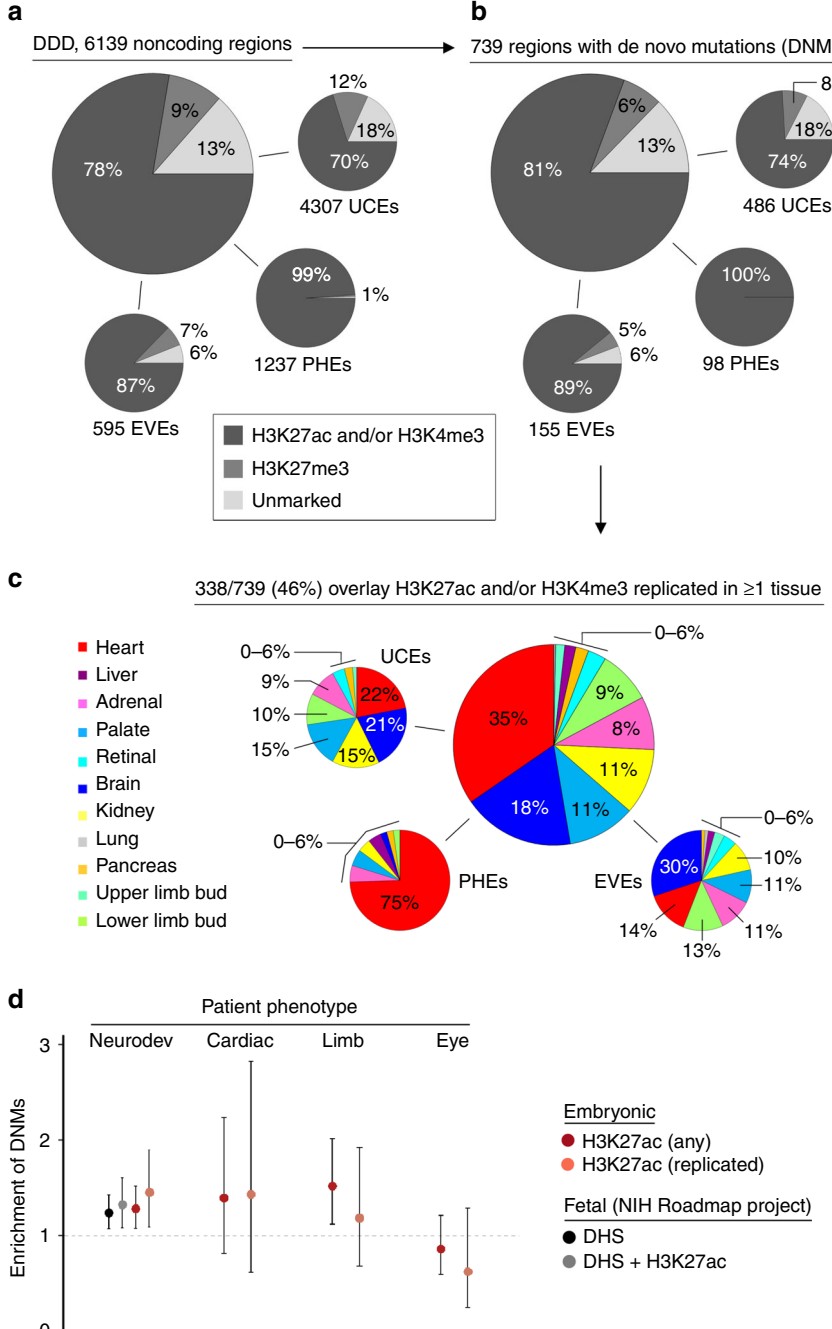

**Fig. 6 Overlay of non-coding de novo mutations linked to developmental disorders. a** The Deciphering Developmental Disorders (DDD) study included 6139 non-coding regions in its sequence analysis of trios comprising affected individuals and unaffected parents[2,32]. These non-coding regions were selected on the basis of high sequence conservation (ultra-conserved elements, UCEs, n = 4307), experimental validation (experimentally validated enhancers, EVEs, n = 595) or identification as a putative heart enhancer (PHE, n = 1237). Overlap with any H3K27ac, H3K4me3 or H3K27me3 1 kb bins is shown as an aggregate and for each individual category (UCE, EVE or PHE). **b** Equivalent overlap is shown for the 739 regions in which disease-associated de novo mutations (DNMs) were identified. **c** In total, 46% of DNM-positive regions were situated (+/− 1 kb) in at least one tissue-replicated H3K27ac and/or H3K4me3 bin. Over half of the disease-associated overlap was covered by the heart/LV (35%) and brain (18%). 75% of the disease-associated PHE regions were situated within 1 kb of a heart/LV-specific histone mark. **d** Enrichment in the number of DNMs overlapping (+/− 1 kb) H3K27ac marks during human organogenesis for individuals with neurodevelopmental (n = 671 cases), cardiac (n = 124 cases), limb (n = 312 cases) and eye (n = 288 cases) phenotypes. The circles represent the observed/expected ratio with asymmetrical error bars showing the 95% confidence limits calculated for a Poisson distribution (http://ms.mcmaster.ca/peter/s743/poissonalpha.html). For the neurodevelopmental phenotypes, this included analysis against DNAse hypersensitivity data and H3K27ac data from second trimester fetal brain[10].

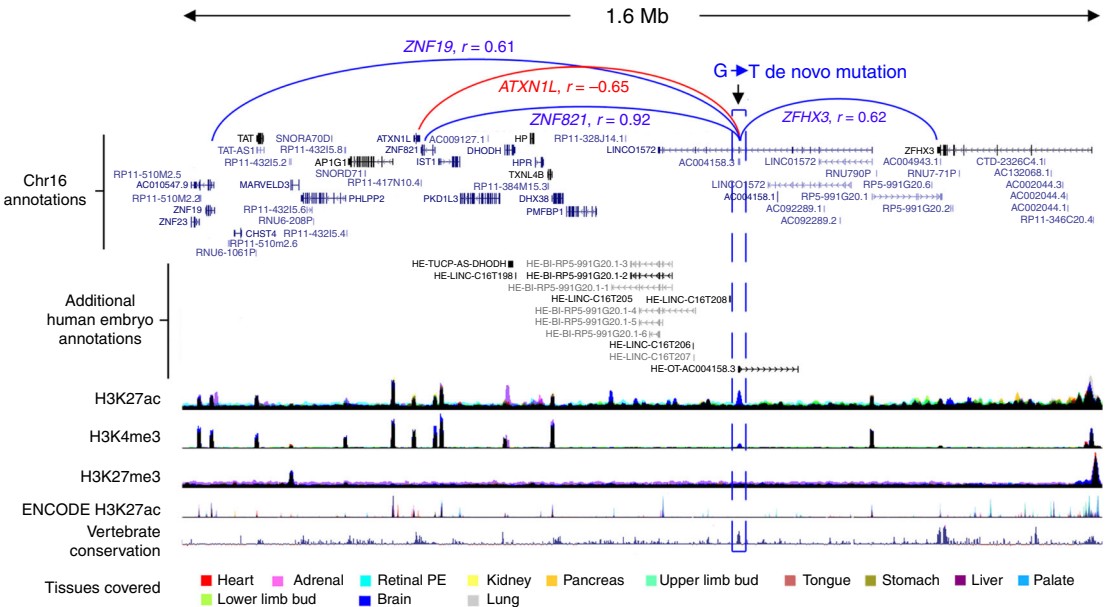

**Fig. 7 Neurodevelopmental de novo mutation within brain-specific histone modification correlated to surrounding gene expression.** An intergenic G-to-T de novo mutation (DNM; hg38, chr16:72427838) is shown for a patient with a neurodevelopmental phenotype. Tracks are shown demonstrating additional human embryonic non-coding transcription (enriched in human embryonic brain), the three epigenomic marks, ENCODE data[26] and conservation amongst vertebrates. The DNM overlaps a brain-specific (dark blue) H3K27ac and small H3K4me3 mark. The highest correlations are shown, notably to the promotors of *ZNF821* ($r = 0.92$) (dark blue) with anticorrelation ($r = -0.65$, red) to the adjacent gene, *ATXN1L*.

promoters reinforced our previous findings based solely on computational analysis of 5′ flanking regions for the importance of NR5A1 in the adrenal and HNF4A in the liver[15]. However, the integrated sampling of numerous sites uncovered far more complex patterns of regulation operating across tissues. The enrichment of *PITX1*-binding motifs in active regulatory regions uniquely shared across the limb bud and palate fits with mutations in *PITX1* causing limb defects and cleft palate[37]. Similarly, GATA4 and GATA6, inferred from regulatory regions shared uniquely between the heart and pancreas, are the only two TFs linked to the dual phenotype of cardiac malformation and monogenic diabetes[38,39]. Overlaying GWAS data with chromosomal conformation studies from older human fetal brain has prioritised target genes for risk of schizophrenia[40]. These techniques are yet to be applied at scale in much smaller human embryonic tissues. However, because we integrated the data from many tissues we can begin to make correlations of enhancer activity to target genes over megabase distances (Fig. 7). While these preliminary correlations are somewhat rudimentary, we have made them available as tracks on UCSC Genome Browser because where linked to expression of the same gene, it might become possible to group individual enhancers into larger clusters to increase statistical power. The latter can be otherwise limiting when causally linking non-coding elements to developmental disorders.

Deciphering profiles of H3K27me3 alongside other regulatory marks and expression profiles was informative. We did not observe bivalent marking of developmental promoters poised for gene expression before reaching the limits of resolution, at which gene sets for structures such as mesenchyme and nerves were common to all tissues. Instead, we discovered that organ-specific developmental programmes were disallowed in other human embryonic tissues by active repression at a series of gene promoters. The ontology of these gene sets, including many encoding TFs, inferred they are an important aspect of ensuring correct cell fate decisions. This realisation opens up an opportunity for more rigorous benchmarking of differentiated hPSCs, including

organoids, both for proximity to the intended lineage in how appropriate gene expression is activated but also against a clearly defined set of epigenomic features for how undesired cell fates are avoided.

In summary, we present an integrated atlas of epigenomic regulation and transcription responsible for human organogenesis. We make all datasets freely available alongside analysis tracks on the UCSC Genome Browser. The uncovering of cryptic regulatory regions and patterns of regulation across organs arose because of direct study of human embryonic tissue. The data complement current international projects such as the Human Cell Atlas[41], by providing greater resolution of regulatory information and depth of sequence information. Moreover, our integrated analyses establish a framework for prioritising and interpreting disease-associated variants discovered by WGS[42] and provide clear routes towards understanding the underlying mechanisms.

## Methods

**Sample dissection.** Human embryonic material was collected under ethical approval from the North West Research Ethics Committee (18/NW/0096), informed consent from all participants and according to the Codes of Practice of the Human Tissue Authority[15]. Tissue collection took place on our co-located clinical academic campus overseen by our research team ensuring immediate transfer to the laboratory. Material was staged by the Carnegie classification, and individual tissues and organs were immediately dissected (Supplementary Data 1 and 2). The material collected here for epigenomic analysis was matched to material isolated for a previous transcriptomic study[15], and the dissection process was identical. In brief, the pancreas, adrenal gland, whole brain, heart, kidney, liver, limb buds, lung, stomach and anterior two-thirds of the tongue were visible as discrete organs and tissues. All visible adherent mesenchymes, including capsular material (adrenal), were removed under a dissecting microscope. The ureter was removed from the renal pelvis. A window of tissue was removed from the lateral wall of the left ventricle of the heart. The dissected segment of liver avoided the developing gall bladder. The trachea was removed where it entered the lung parenchyma. The stomach was isolated between the gastro-oesophageal and pyloric junctions. The palatal shelves were dissected on either side of the midline. The eye was dissected, and the RPE peeled off mechanically from its posterior surface (facilitated by the dark pigmentation of the RPE allowing straightforward visualisation).

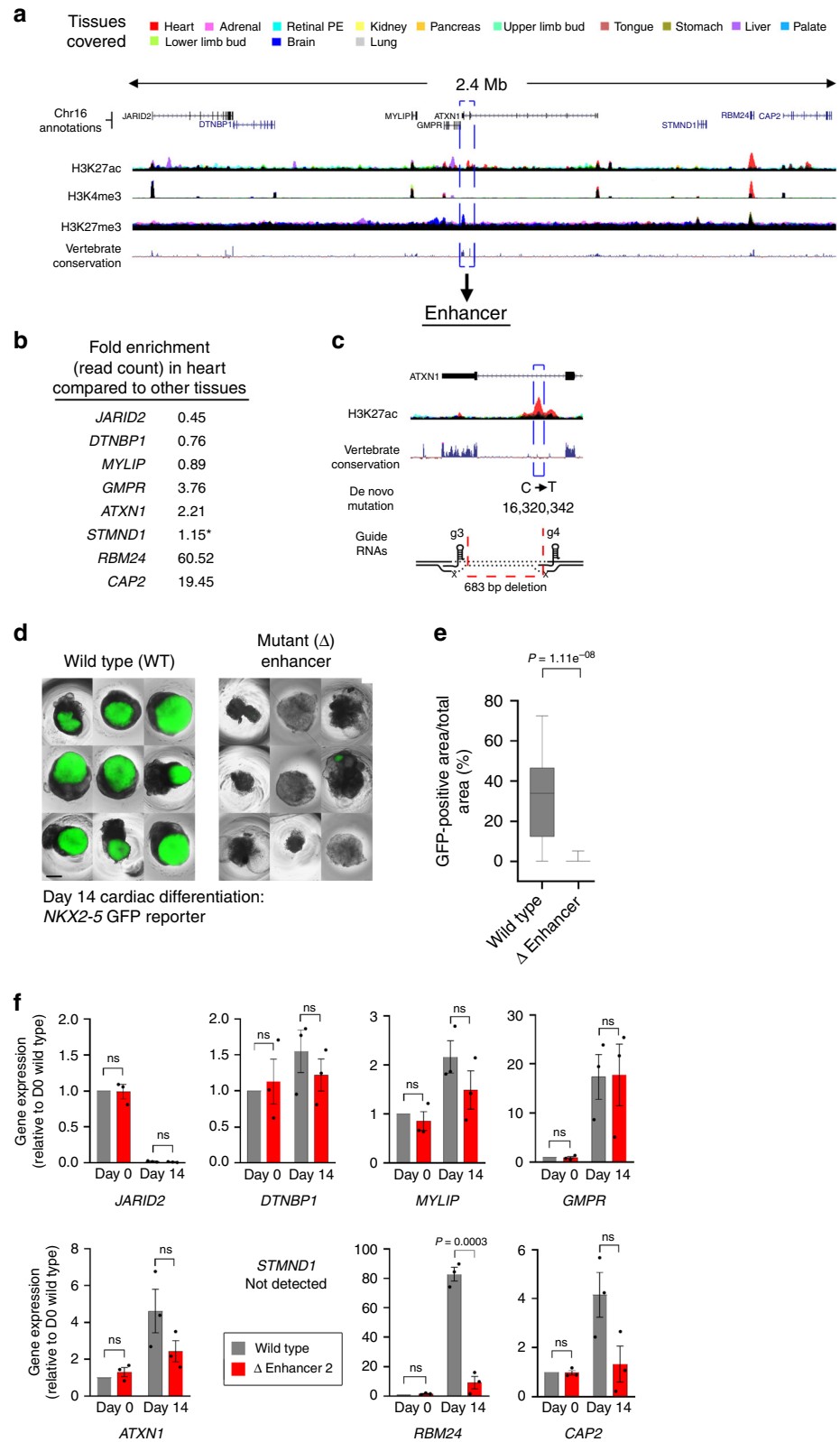

**a** Tissues covered: ■ Heart ■ Adrenal ■ Retinal PE ■ Kidney ■ Pancreas ■ Upper limb bud ■ Tongue ■ Stomach ■ Liver ■ Palate ■ Lower limb bud ■ Brain ■ Lung

2.4 Mb

Chr16 annotations: JARID2, DTNBP1, MYLIP, ATXN1, GMPR, STMND1, RBM24, CAP2

H3K27ac
H3K4me3
H3K27me3
Vertebrate conservation

Enhancer

**b** Fold enrichment (read count) in heart compared to other tissues

| | |
|---|---|
| JARID2 | 0.45 |
| DTNBP1 | 0.76 |
| MYLIP | 0.89 |
| GMPR | 3.76 |
| ATXN1 | 2.21 |
| STMND1 | 1.15* |
| RBM24 | 60.52 |
| CAP2 | 19.45 |

**c** ATXN1

H3K27ac
Vertebrate conservation
De novo mutation   C→T   16,320,342
Guide RNAs   g3   g4   683 bp deletion

**d** Wild type (WT)    Mutant (Δ) enhancer

Day 14 cardiac differentiation: NKX2-5 GFP reporter

**e** P = 1.11e⁻⁰⁸
GFP-positive area/total area (%)
Wild type / Δ Enhancer

**f** Gene expression (relative to D0 wild type)

JARID2 — ns (Day 0, Day 14)
DTNBP1 — ns
MYLIP — ns
GMPR — ns
ATXN1 — ns
STMND1 Not detected
RBM24 — P = 0.0003
CAP2 — ns

■ Wild type   ■ Δ Enhancer 2

Tissues were gently teased apart before cross-linking in 1% formaldehyde for 10 min at room temperature. Fixation was quenched with 125 mM glycine for 5 min at room temperature before centrifugation, removal of the supernatant and washing twice with 1 ml of PBS. The final PBS supernatant was discarded and samples stored at −80 °C until use (Supplementary Data 1). The sex of tissue from embryos was determined by PCR for X and Y chromosome-specific primers (Supplementary Data 9).

**Chromatin immunoprecipitation (ChIP), RNA isolation and sequencing.** All ChIPseq datasets were in biological replicate, except for the stomach and tongue (Supplementary Data 1). Each sample was placed in lysis buffer [10 mM HEPES, 0.5 mM EGTA, 10 mM EDTA, 0.25% Triton X-100 and protease inhibitor cocktail (Roche)] on ice for 5 min, and nuclei released with ten strokes in a Dounce homogeniser. Nuclei were pelleted by centrifugation at 700 rcf for 10 min at 4 °C, and the supernatant discarded. Nuclei were resuspended in ice cold wash buffer

**Fig. 8 Deletion of enhancer upstream of *RBM24* disrupts cardiomyocyte differentiation. a** Schematic of a 2.4 mb locus on chromosome 6 containing eight protein-coding genes centred on a cardiac-specific H3K27ac peak (red, broken line box) within the last intron of *ATXN1*. This enhancer harbours a DNM from the DDD cohort associated with congenital heart disease[2]. The histone modification tracks contain datasets from all the colour-coded tissues. **b** Fold enrichment in the heart/LV dataset compared with the average across all other tissues for RNAseq read counts of the genes shown in (**a**). *RBM24* and *CAP2* are considerably enriched in the human embryonic heart. **c** Magnified schematic of the enhancer shown in (**a**) showing the location of the DNM and the CRISPR-Cas9 approach for deletion. **d** EBs from wild-type and enhancer deletion (mutant) hPSCs containing the *NKX2-5*-GFP reporter. The images showing nine EBs for wild-type and mutant are after 14 days of the cardiomyocyte differentiation protocol[34,35]. Size bar, 500 μm. **e** Box and whisker plot (box showing 25th−75th percentile and median line with min−max as whiskers) quantifying GFP across all wild-type ($n = 29$) and mutant EBs ($n = 30$). **f** RT-qPCR for expression of all the protein-coding genes across the 2.4 mb locus depicted in (**a**). Ten different clones were used in three independent experiments for mutant EBs with ten, ten and nine clones for wild-type control. Error bars represent S.E.M. from the three independent differentiation experiments (the individual dots). Time points for RT-qPCR were day 0 (undifferentiated hPSCs) and day 14 (cardiomyocyte progenitors[34,35]). While *CAP2* expression appeared reduced, *RBM24* was the only gene with significantly lowered expression following deletion of the cardiac-specific enhancer shown in a). Significance was assessed using a two-tailed Student's *t* test (ns, not significant).

(10 mM HEPES, 0.5 mM EGTA, 1 mM EDTA, 20 mM NaCl and protease inhibitor cocktail), then pelleted by centrifugation at 700 rcf for 10 min at 4 °C and the supernatant discarded. Nuclei were lysed (50 mM Tris-HCl, 10 mM EDTA, 1% SDS and protease inhibitor cocktail) and sonicated under prior optimised conditions (Diagenode Bioruptor). Sufficient sample was prepared to allow in parallel immunoprecipitation for H3K4me3, H3K27ac and H3K27me3 to minimise technical variation. In total, 1 μg of DNA equivalent was used for each pulldown. Samples were diluted with nine volumes of dilution buffer (16.7 mM Tris-HCL, 1.2 mM EDTA, 167 mM NaCl, 0.01% SDS and 1.1% Triton X-100). In all, 20 μl of ChIP-grade magnetic beads were washed twice in dilution buffer, and incubated with each sample for 3 h on a tube rotator at 4 °C to preclear the sample. The beads were separated, and the pre-cleared lysate transferred to a separate tube. The magnetic bead pellet was discarded. For each histone modification, 3 μg of antibody (Supplementary Data 2) were added to each sample followed by incubation on a tube rotator at 4 °C overnight. In all, 30 μl of magnetic beads were washed twice in immunoprecipitation dilution buffer and incubated with samples for 3 h at 4 °C. Beads were collected and washed twice with wash buffer A (20 mM Tris-HCl, 2 mM EDTA, 50 mM NaCl, 0.1% SDS and 1% Triton X-100), once with wash buffer B (10 mM Tris-HCl, 1 mM EDTA, 250 mM LiCl, 1% NP40 and 1% deoxycholate) and twice with TE buffer (10 mM Tris-HCl and 1 mM EDTA). Beads were then incubated in elution buffer (1% SDS and 100 mM NaHCO₃) for 30 min at 65 °C and the beads discarded. The resulting samples were incubated with 167 mM NaCl for 5 h at 65 °C to remove crosslinks followed by 1 h incubation with 14 μg of proteinase K. The resulting chromatin was purified (MinElute, QIAGEN).

DNA libraries were constructed according to the TruSeq® ChIP Sample Preparation Guide (Illumina, Inc.). Briefly, sample DNA (5–10 ng) was blunt-ended and phosphorylated, and a single "A" nucleotide added to the 3′ ends of the fragments in preparation for ligation to an adapter with a single base "T" overhang. Omitting the size-selection step, the ligation products were then PCR-amplified to enrich for fragments with adapters on both ends. The final purified product was then quantitated prior to cluster generation on a cBot instrument (Illumina). The loaded flow cell was sequenced (paired-end) on a HiSeq2500 (Illumina). In total, ChIPseq was carried out in three batches with hierarchical clustering analysis to examine for batch effect (Supplementary Fig. 7).

RNAseq for this study has been described previously (Supplementary Data 3)[15]; using identical methodology, we added single datasets for pancreas and tongue and two datasets for lung to create biological transcriptomic replicates for all tissues (Supplementary Data 3).

**Mapping of ChIPseq and RNAseq.** The first batch of ChIPseq was mapped originally to hg19 using Bowtie 1.0.0 (parameters -m1 -n2 -l28, uniquely mapped reads only)[43] and peaks called using MACS2 (2.0.10.20131216)[44] against a common input dataset (derived from all tissues). MACS parameters used were as follows: band width 300 bp, mfold 5–50 (used in cross-correlation for fragment length estimation), q-value cut-off 0.05. To prioritise candidate enhancers for transgenic testing, H3K27ac data from ENCODE (seven cell lines) and NIH Roadmap (154 samples)[10,26] were mapped similarly. Subsequently, all data, including the external H1 hPSC and adult pancreas data (Fig. 3c), were mapped to hg38 using STAR (2.4.2a)[45]. ChIPseq reads were trimmed to 50 bp for consistency, and only uniquely mapped reads were retained. For ChIPseq, spliced mappings were suppressed by setting the parameter "alignIntronMax" to 1. The full STAR parameters for ChIPseq were as follows: "–alignIntronMax 1,–seedSearchStartLmax 30,–outSAMattributes All, and–outSAMtype BAM SortedByCoordinate". GENCODE 25 gene annotations were used for RNAseq mapping and read counting[46]. The full STAR parameters for RNA-seq were as follows: "—outSAMattributes All,–quantMode GeneCounts –out, SAMtype BAM SortedByCoordinate".

**Chromatin and promoter state analysis.** Genomic segmentation was performed using chromHMM (version 1.11)[17] under default parameters labelling samples by tissue and histone modification. The three histone marks allowed for eight segment states.

Clustered promoter states were identified for an annotated set of 19,791 protein-coding genes in each tissue using ngs.plot k-mean clustering (version 2.61) on unnormalized reads for the combined dataset of replicated RNA-seq and ChIPseq for H3K4me3, H3K27ac and H3K27me3[21]. Default settings allowed for five clusters based on rank profiles of read counts 3 kb either side of the TSS. The returned clusters were then classified according to characteristics detected in both replicates into five major promoter states (Fig. 2): actively repressed (H3K27me3 signal >50% of maximum and mean transcript counts <10% of maximum); Narrow expressed [H3K4me3 signal >25% of maximum with >90% of reads downstream of the TSS and skew >0.65 (measured across 100 equidistant percentiles from TSS to +3 kb); and mean transcript counts >10% of maximum]; broad expressed (as for narrow expressed, but with skew <0.65); bidirectional expressed (H3K4me3 signal >25% of maximum with <90% of reads downstream of the TSS; and mean transcript counts >10% of maximum); Bi-dir2 (as for bidirectional expressed, but without the H3K4me3 signal); Expressed2 (H3K4me3 signal >25% of maximum with mean transcript counts <10% of maximum); and inactive (<25 of maximum for H3K4me3 and H3K27me3 and mean transcript counts <10% of maximum). This approach left each gene uniquely assigned to one cluster in any tissue. Bi-dir2 was only identified in RPE (Supplementary Fig. 2). Expressed2 was detected in the lung, liver and brain (Supplementary Fig. 3). While superficially this category lacked significant transcription, in fact, total gene-level read counts were very similar to Broad expressed. However, longer mRNA and longer first introns limited transcript detection at the TSS (Supplementary Fig. 4). The full listings are in Supplementary Data 4. The over-representation of TFs in the TSS regions marked with H3K27me3 and featuring CpG islands was assessed on the dataset of 1659 genes encoding all the TFs compared against the remaining 18,132 non-TF genes using Fisher's exact test (two-sided). To search explicitly for bivalency of H3K4me3 and H3K27me3 at gene promoters, the default parameters of ngs.plot were extended to allow more clusters (7, 10 or 11) as described in the "Results". This generated a sub-category of H3K27me3 for each tissue that also contained H3K4me3 (Supplementary Figs. 6, 8 and 10).

Alluvial plots were created using the R package Alluvial Diagrams version 0.2-0[47] with modification of the R code to reorder the horizontal splines (alluvia) within each tissue to keep similar colours together.

**Annotation set enrichment for genes and genomic regions.** Lists of genes from the associated promoter state were tested for enrichment of annotations using the xEnricherGenes function from the R package XGR version 1.1.1 under default parameters[48]. Background comprised all remaining annotations used in the ngs.plots either for embryonic tissues (Fig. 3a, b) or from hPSCs through to adult pancreas (Fig. 3c, d). Pathway annotation and gene set enrichment analysis for the subset of H3K27me3/H3K4me3 dual-marked promoters in each tissue was undertaken using ReactomePA (release 3.10) under default parameters, which associates genes to their known functions based on the REACTOME pathway database[49].

**Transgenic analysis in zebrafish.** A systematic approach identified candidate enhancers that were human embryo-enriched and tissue-specific. We identified marks from the first batch of H3K27ac with RPKM ≥ 25 and ≥2.5-fold enrichment in the human embryo compared to ENCODE (7 cell lines)[26] or NIH Roadmap datasets (154 tissues, including fetal datasets from the second trimester)[10]; and that were undetected in the FANTOM5 project[27]. To filter these embryonic marks for tissue specificity, an initial dataset was selected at random and peaks called that were >200 bp. The H3K27ac datasets from other embryonic tissues were then overlaid sequentially in random order. Only called peaks >200 bp were included. After each addition, any peaks with <50% overlap between the new and existing dataset were retained. For those retained regions, overlapping sequence was filtered out. Once completed, the final set of human embryo-enriched, tissue-specific sequences were again filtered for regions >200 bp. Re-running the tissue-specificity algorithm for random addition of datasets resulted in a 99.6% match to the first analysis. These candidate enhancer regions were filtered for sequence conservation (PhastCons LOD score >50)[50] and correlated with surrounding transcription (≤1 mb in either direction). We manually inspected the remainder for proximity (<1

mb) to genes encoding TFs associated with major developmental disorders and ensured no H3K4me3 or polyadenylated transcription in the immediate vicinity (i.e., an unannotated promoter). This resulted in 44 candidate enhancers from which we tested ten. The candidate sequences were first cloned in TOPO vector using pCR8/GW/TOPO TA cloning kit (cat. no. K252020, Invitrogen Thermo Fisher Scientific) and then recombined into the reporter vector Minitol2-GwB-zgata2-GFP-48[51] using the Gateway LR clonase II Enzyme mix (cat. no. 11791020, Invitrogen Thermo Fisher Scientific). The reporter vector contains a robust mid-brain enhancer as an internal control for transgenesis.

Transgenic fish were generated with the Tol2 transposon/transposase method of transgenesis[52]. *Danio rerio* embryos were collected from natural spawning and injected in the yolk at the one-cell stage. The injection mixture contained 50 ng/µl Tol2 transposase mRNA, purified enhancer test vector and 0.05% phenol red. The concentration of the enhancer test vector was between 15 and 30 ng/µl. Injected embryos were visualised from 24 hpf to 48 hpf in an Olympus stereomicroscope coupled to a fluorescence excitation light source in order to detect the pattern of GFP.

Embryos and adults zebrafish were maintained under standard laboratory conditions. They were manipulated according to Spanish and European regulation. All protocols used have been approved by the Ethics Committee of the Andalusian Government (license numbers 450-1839 and 182-41106 for CABD-CSIC-UPO).

**Genome binning, normalisation and thresholding**. The genome was parsed into 3,087,584 non-overlapping contiguous 1 kb bins to compare ChIPseq profiles across tissues and replicates. Reads were counted into bins according to their mapped start position using csaw (version 3.11)[53]. Reads from mitochondrial and unplaced chromosome annotations were removed. A further 697 bins were filtered out for possessing >10,000 reads in all samples or if the mean read count from input controls was ≥50% of the mean read count of all samples or for being situated in pericentromeric regions (using table ideogram from UCSC; listed in Supplementary Data 8). For Pearson correlations with the surrounding transcription binned, read counts were downsampled statistically using subSeq[54] weighting each sample by the value of the 99th percentile.

Downsampling of read counts to the 99th percentile was used to generate the custom elbow threshold that called bins as marked or not for subsequent downstream analyses. When read counts were ordered and plotted by rank, the resulting graph was typically exponential with most bins having zero or very few reads (below the elbow threshold) and a small number of bins with very high read counts (above the elbow threshold). The elbow was defined as the point on the line with the shortest Euclidian distance to the maximum rank intercept with the $x$ axis. Our code (arseFromElbow) to determine these thresholds from a vector of counts is available on github[55]. Phi correlation was used to measure the agreement between tissue replicates called by the 1 kb binning method compared to MACS[44]. Hierarchical clustering of datasets was undertaken to assess potential batch effect (displayed by heatmap) based on the combined set of the 10,000 most highly ranked bins from each sample. Sets of tissue-specific (replicated in exactly one tissue) and tissue-selective bins (replicated in a given tissue and up to a half of all samples) were produced for each embryonic tissue. EulerGrids showing pattern frequencies of bins across samples were produced using the function plotEuler[56] as an adaptation of a proposal from Reynolds and colleagues[57] on Biostars.org[58].

**Motif analysis**. HOMER v4.9 was used to search for enriched motifs in selected sets of bins[59]. For selected 1 kb bins marked with H3K27Ac, the background set was the remainder of bins with replicated H3K27Ac across all tissues ($n = 160,043$).

**hPSC culture, cardiomyocyte differentiation and enhancer inactivation**. *NKX2-5*eGFP/w hPSCs (*NKX2-5*-GFP reporter) were maintained on mouse embryonic fibroblasts and passaged using TrypLE Select (Thermo Fisher Scientific) with cardiomyocyte differentiation performed via embryoid body (EB) formation[34,60,61]; except for modified concentrations of BMP4 (25 ng/ml) and Activin A (25 ng/ml). Day 14 EBs were imaged for GFP and then collected into Trizol Reagent (Thermo Fisher Scientific) for RNA extraction according to the manufacturer's specifications followed by treatment with DNaseI Kit (Sigma). First-strand cDNA synthesis and qPCR were performed using the Tetro cDNA synthesis kit (Bioline) and Power Up Sybr Green (Invitrogen). Primers for genes in the *RBM24* locus are in Supplementary Data 9.

CRISPR-Cas9 guide-RNAs were designed to the cardiac-specific enhancer in the final intron of *ATXN1* using CHOPCHOP[62] version 3 (Supplementary Data 9). Ribonucleoprotein (RNP) complexes targeting flanking sites around the enhancer locus were made and cotransfected into hPSCs[61]. Cells were cloned by serial dilution and screened for enhancer deletion by PCR (Supplementary Data 9). PCR screening was carried out using Q5 polymerase (NEB) with GC enhancer (98 °C 180 s) 35 cycles of (98 °C 20 s, 62 °C 30 s, 72 °C 90 s), 98 °C 180 s) to confirm presence of wild-type (1069 bp), heterozygous (401 bp) or deleted (438 bp) amplicons in clones.

EBs were imaged on an Olympus IX83 inverted microscope and captured using an Orca ER camera (Hamamatsu) through CellSens software (Olympus). Images were processed and analysed using Fiji ImageJ with GFP quantified as a proportion of the total EB area. Montages for bright-field and FITC channels were created

consistently for all EBs using a Gaussian Blur with sigma radius of 5:00, and a colour threshold:brightness lower limit of 52:255. Regions of interest were mapped and the relative GFP-positive area quantified. Significance was assessed using two-tailed Student's $t$ test.

**Reporting summary**. Further information on research design is available in the Nature Research Reporting Summary linked to this article.

## Data availability

The authors declare that all data supporting the findings of this study are available within the article and its supplementary information files or from the corresponding author upon reasonable request. ChIPseq and RNA-seq datasets have been deposited in the European Genome Phenome repository [www.ebi.ac.uk/ega/home] under accession codes: EGAS00001003738 (RNA-seq), EGAS0001004335 (ChIPseq) and EGAS00001003163 (study). Supplementary Data 1–3 detail the human embryonic material contributing to these datasets. To view data in the UCSC genome browser, a trackhub is at http://www.humandevelopmentalbiology.manchester.ac.uk/. The external databases were used by the study: ENCODE [www.encodeproject.org/][26]; NIH Roadmap [www.roadmapepigenomics.org/][10]; FANTOM5 [fantom.gsc.riken.jp/5/][27]; xEnricherGenes (from XGR v1.1.1)[48]; Reactome pathway database [reactome.org/][49]; and HOMER v4.9 [homer.ucsd.edu/homer/][59].

## Code availability

The following codes are freely available[55,56]: arseFromElbow [https://rdrr.io/github/davetgerrard/utilsGerrardDT/src/R/arseFromElbow.R] and plotEuler [https://github.com/davetgerrard/utilsGerrardDT].

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

## Acknowledgements

We are very grateful to all women who consented to take part in our research programme and for the assistance of research nurses and clinical colleagues at the Manchester University NHS Foundation Trust. We thank Peter Briggs and Andy Hayes of the Bioinformatics and Genomic Technologies Core Facilities at the University of Manchester. The work was supported by Wellcome grants 088566, 097820 and 105610, with additional support from MRC project grants MR/L009986/1 to N.B. and N.A.H., MR/J003352/1 to K.P.H., and MR/000638/1 and MR/S036121/1 to N.A.H. R.E.J. was an MRC clinical research training fellow, and S.J.W. was an MRC doctoral account PhD student. J. L.G.S. was supported by the Marató TV3 Fundacion (Grant No. 201611).

## Author contributions

D.T.G., A.A.B and N.A.H. devised the study and planned experiments. K.P.H., M.B., S.J.W., R.E.J., A.D.S. and N.B. were involved in study design and oversight of human embryonic material collection (N.A.H., R.E.J. and K.P.H.). A.A.B. processed the human embryonic material and prepared samples for all sequencing analyses. D.T.G., I.D. and P.Z. conducted the bioinformatics analyses. J.L.G.S., S.J.G. and P.N.F. undertook the transgenic analyses. N.A.H., D.T.G., P.S. and M.E.H. conducted the analysis of developmental disorders. M.J.B. and M.G.G. conducted the hPSC analyses. D.T.G. and N.A.H. wrote the paper with input from A.A.B. and editing from K.P.H., N.B., A.D.S. and J.L.G.S. N.A.H. is the guarantor.

## Competing interests

The authors declare no competing interests.
