## [Peer Review File · Nature Communications]

Reviewers' Comments:

Reviewer #1:

Remarks to the Author:

Dear authors,

The manuscript "The epigenomic landscape regulating organogenesis in human embryos linked to developmental disorders" by Gerrard et al. provides unique insight regarding epigenetic landscape dynamics of H3K27Ac, H3K4me3 and H3K27me3 during a short period of human early development (CS14-CS22). It is very rare to have access to these early stages of human development. Moreover, the combination of human material accessibility and the epigenetic analysis used is unique to the Hanley lab and provides valuable information regarding this subject.

Having said that, I think the paper would benefit from a clarification on the exact number of embryos used in a separate table mentioning what organs belong to each embryo and were used in what Chip-seq batch.

Most importantly, I could not find information regarding the sex of the embryos used. This information is super important regarding the epigenetic marks, particularly H3K27me3, on the silent X chromosome in female embryos as that differs from male embryos. The embryos should be stratified by sex and the epigenetics of genes in the X chromosome analysed separately. If this is not possible, the sex of each embryo should be at least stated for each organ/batch/Chipseq. And regardless of whether it is possible to separately analyse the 3 epigenetic marks in males and females, perhaps the epigenetic analysis of the genes on X chromosome should be considered separately from the autosomes not to confound results.

Reviewer #2:

Remarks to the Author:

In their manuscript, Gerrard et al. report the generation of a new dataset, profiling three different histone modifications across multiple organs during human organogenesis. This is a very useful resource that sheds light into a period of human development that remains understudied. The authors integrate this new dataset with previously generated RNA-seq data in order to explore the regulatory landscape of human organogenesis and reach several interesting conclusions: They cluster promoters according to the assayed features, finding limited evidence for the presence of bivalent promoters, identify differences in the mode of regulation between developmental transcription factors and organ-specific effector genes, and finally attempt to link the newly identified regulatory regions to human developmental disorders.

Major concerns

While the dataset is definitely impressive and will undoubtedly be very useful to the community, I find that the manuscript could benefit from more rigorous analyses that would consolidate some of the claims presented by the authors. Many of these claims are currently supported by limited and sometimes weak analyses. Specifically:

1. In the first section focusing on promoters, the authors conclude that there is little evidence for promoter bivalency during human organogenesis. This is primarily based on their clustering analysis of promoter states. There are several concerns regarding this analysis:
 - a. Clustering was performed using a tool primarily oriented towards visualization and under default settings (more information should be provided on the precise parameters but in all available options the number of clusters seems to be determined by the user). Where is the evidence that the data can be optimally described by five clusters? Would it be possible that allowing for more clusters would reveal bivalent promoters as a subset of the repressed promoters (especially given the faint H3K4me3 signal observed in the repressed state)?

b. This concern becomes even more relevant if one takes a closer look into Figure 3a. Actively repressed promoters clearly show a bimodal behaviour. Some remain repressed across all assayed tissues whereas others switch between the expressed and repressed state. Could this group correspond to bivalent promoters? An easy way to check would be to split the two groups (globally repressed vs facultative repressed and compare the H3K4me3 signal between the two groups).

c. Furthermore, since bivalent promoters are primarily described based on the co-occurrence of H3K4me3 and H3K27me3, it might be informative to additionally perform the clustering only considering these two modifications (and allowing for four or more clusters). H3K27ac and active transcription are not expected to be found in bivalent promoters so they are not particularly informative for this particular question but their presence might distort the clustering analysis.

2. In the last part of the manuscript, the authors intersect their putative enhancer annotation with a set of non-coding regions previously identified to be involved in developmental disorders. They then show that ~50% of noncoding de novo mutations fall within an active chromatin region. This is not surprising as these regions were previously selected for being conserved or for showing evidence of enhancer activity (experimentally validated or putative enhancers). A key step towards associating these mutations with the respective developmental phenotype would be to show that they are active in the organ that is most relevant to the respective disorder (i.e. neuronal organ for neurodevelopmental disorders and heart for heart defects). Surprisingly, with the exception of the single anecdotal case presented in Figure 7a, the authors don't attempt to associate organ-specific regulatory activity with the observed phenotype for each DNM. Even in Figure 6d it seems that they test for enrichment of their entire H3K27ac set against different phenotypes (although I cannot be sure as this analysis is not properly documented in the methods).

3. The authors should provide more information regarding their ChIP-seq to RNA-seq correlation analysis. This is briefly mentioned in the results section and is almost entirely absent from the methods. Which correlation metric was used? Did the authors correct for the effect of spurious correlations due to multiple testing (for example by shuffling regulatory regions and genes from distant chromosomes)? How was the significance of the observed correlations estimated? Shouldn't the results of this analysis be included in the Supplementary Material? This analysis cites Figure 7, which only presents data for one gene!

Minor comments

1. It's quite alarming that some high rank patterns from the binary activity calling arise from combinations between single replicates of different organs (i.e. Fig. 5b upper limb/pancreas ranks 6th, adrenal and heart is 8th). Do the authors attribute this to the biological samples or to the approach used for calling activity? Could it also be related to differences in the developmental stages of the samples (i.e. clustering of samples from different tissues but more similar developmental stages)?

2. Regarding the transgenic zebrafish experiment: It's not clear how sequence conservation in zebrafish was tested. Was this based on sequence only or was synteny also considered? The authors should state this in the methods. Scoring sequence conservation of non-coding elements over so long evolutionary distances is a difficult task and therefore absence of evidence for conservation should not be interpreted as evidence for its absence. Furthermore there is some ambiguity in the terms used by the authors (i.e. novel for previously unannotated, which could be confused with evolutionary novel; distance for evolutionary distance, which in the context of enhancer function could be confused with physical distance). I would therefore recommend rephrasing the concluding statements as something similar to this: "These data imply that our H3K27ac detection marks previously unannotated human enhancers, which function over considerable evolutionary distance despite the absence of detectable sequence conservation. This is likely associated with the overall conserved TF activity between homologous organs (Stergachis et al. Nature 2014)".

3. GO enrichments in temporal analysis: Not clear what the background is but it seems that all

19,700 genes in Figure 3a were used. Since the point of the analysis is to identify differences between genes derepressed in the transition from pluripotency to organogenesis vs from organogenesis to adult organ, it's more appropriate to limit the background set to genes expressed in the pancreas during any of the three stages. This might improve the resolution and remove confounders associated with general expression patterns in the pancreas (ex. involvement in diabetes).

4. Overall, more information should be added to the methods section. The most critical issues were already highlighted above, but the authors could include more information in many sections (ex. parameters for MACS, STAR, read counting method, enrichment tests etc).

Reviewer #3:

Remarks to the Author:

In this manuscript, Gerrard and colleagues mapped landscapes of histone H3K4me3, H3K27ac and H3K27me3 in thirteen tissues during human organogenesis. They classified discrete promoter states and identified a number of putative human embryo-enriched intergenic enhancers, several of which were validated in zebrafish. They further compared the promoter states and H3K27ac patterns across tissues. In addition, their analysis linked histone modification marked regions to de novo mutation sites which are associated with human developmental disorders.

Characterizing the epigenomic patterns during organogenesis is important for understanding the mechanisms of human organ development in health and disease. However, the author's conclusions on the epigenome analysis of the entire organ are not convincing, and such analysis did not provide useful information and insights into our understanding of human organ development. For example, the fetal liver contains > 20 cell types, including endoderm-derived cells and mesoderm-derived cells. These mixed ChIP-seq signals are not suitable for identifying cell type-specific DNA regulatory sequences. The human epigenetic atlas should be established at the cell lineage level or at the single cell level, but not at the organ level.

Your manuscript entitled "The epigenomic landscape regulating organogenesis in human embryos linked to developmental disorders" has now been seen by three referees, whose comments are appended below. You will see from their comments copied below that while they find your work of considerable potential interest, they have raised quite substantial concerns that must be addressed. In light of these comments, we cannot accept the manuscript for publication, but would be interested in considering a revised version that addresses these serious concerns.

We are very grateful for the time of the editor and reviewers. The reviewers make a number of helpful points, which we have incorporated to improve our manuscript. We are very pleased that our work is of considerable potential interest and very happy to address the points below and in the revised manuscript.

We hope you will find the referees' comments useful as you decide how to proceed. In particular, please include more organ level data (in line with comments from Referee #2); provide much more detailed methods and also, please discuss the limitations of the work – as outlined by Referee #3. Whilst we appreciate the last comment from Referee #3 would be a valuable extension of this work, namely to “establish a human epigenetic atlas at the single cell level not organ level” we understand that is not within the scope of the current paper. As I am sure you are aware, irrespective of whether new data are required to address a reviewer query, we would expect you to address all points in detail in your response to the reviewers/a revised manuscript (if appropriate).

We have provided more organ-level detail, including new analyses using all replicated datasets and new proof of principle on cardiac differentiation from human pluripotent stem cells to exemplify the utility of the overarching dataset (Reviewer #2). We have added more discussion on the limitations of the data (Reviewer #2 and #3). Thank you for understanding that single cell analysis is beyond the scope of the current paper.

Should further experimental data or analysis allow you to address these criticisms, we would be happy to look at a substantially revised manuscript. However, please bear in mind that we will be reluctant to approach the referees again in the absence of major revisions. If the revision process takes significantly longer than six months, we will be happy to reconsider your paper at a later date, as long as nothing similar has been accepted for publication at Nature Communications or published elsewhere in the meantime.

We were keen to return to the journal as soon as possible so that we can make the datasets freely available. Generating the new data, especially human pluripotent stem cell modelling of cardiac gene regulation, has taken three months. As a large confounder, the corresponding author is back in full time clinical NHS duty because of the COVID19 pandemic, hence the delay over the last month or so.

We are committed to providing a fair and constructive peer-review process. Do not hesitate to contact us if you wish to discuss the revision or if there are specific requests from the reviewers that you believe are technically impossible or unlikely to yield a meaningful outcome.

Thank you. We have discussed any relevant points here so that everyone can view it.

When resubmitting your paper, please highlight all changes in the manuscript text file. We also ask that you ensure that your manuscript complies with our editorial policies.

We have tried our best to undertake this.

Reviewers' comments:

Reviewer #1 (Remarks to the Author):

Dear authors,

The manuscript “The epigenomic landscape regulating organogenesis in human embryos linked to developmental disorders” by Gerrard et al. provides unique insight regarding epigenetic landscape dynamics of H3K27Ac, H3K4me3 and H3K27me3 during a short period of human early development (CS14-CS22). Is very rare to have access to these early stages of human development. Moreover, the combination of human material accessibility and the epigenetic analysis used is unique to the Hanley lab and provides valuable information regarding this subject. Having said that, I think the paper would benefit from a clarification on the exact number of embryos used in a separate table mentioning what organs belong to each embryo and were used in what Chip-seq batch. Most importantly, I could not find information regarding the sex of the embryos used. This information is super important regarding the epigenetic marks,

particularly H3K27me3, on the silent X chromosome in female embryos as that differs from male embryos. The embryos should be stratified by sex and the epigenetics of genes in the X chromosome analysed separately. If this is not possible, the sex of each embryo should be at least stated for each organ/batch/ChIPseq. And regardless of whether it is possible to separately analyse the 3 epigenetics marks in males and females, perhaps the epigenetic analysis of the genes on X chromosome should be considered separately from the autosomes not to confound results.

Thank you for the kind comments and the opportunity to update the manuscript. Some embryo metadata were in the Supplementary tables (which we think were available to the Reviewers depending on how the manuscript was transferred from BioRxiv?). We have now ensured that the sex of each embryo is listed by creating a system of anonymised IDs in Supplementary table 1, which allows tracking across samples in the publicly available datasets. Importantly, all replicated datasets in all organs contained both male and female tissue. This mitigates very strongly against the risk of any organ-level differences due to tissue sex as such differences would have been filtered out of downstream analyses as unreplicated information. This most likely explains why we have not seen any clues for confounding between sex chromosomes and autosomes. While we were previously clear about the need for pooling material because of tiny size and the rare nature of the material (as noted and complimented by the Reviewer—thank you), Reviewer #1 has prompted us to add extra detail (e.g. in the Discussion) to emphasise the above.

In summary, we are very grateful for the recognition that our data are rare and valuable. We trust the above response and updated manuscript provides the clarifications that the Reviewer was seeking.

Reviewer #2 (Remarks to the Author):

In their manuscript, Gerrard et al. report the generation of a new dataset, profiling three different histone modifications across multiple organs during human organogenesis. This is a very useful resource that sheds light into a period of human development that remains understudied. The authors integrate this new dataset with previously generated RNA-seq data in order to explore the regulatory landscape of human organogenesis and reach several interesting conclusions: They cluster promoters according to the assayed features, finding limited evidence for the presence of bivalent promoters, identify differences in the mode of regulation between developmental transcription factors and organ-specific effector genes, and finally attempt to link the newly identified regulatory regions to human developmental disorders.

Thank you for these very positive comments and for recognising so clearly, above and below, the value of our datasets to the wider audience.

Major concerns

While the dataset is definitely impressive and will undoubtedly be very useful to the community, I find that the manuscript could benefit from more rigorous analyses that would consolidate some of the claims presented by the authors. Many of these claims are currently supported by limited and sometimes weak analyses.

Reviewer #2's comments have been very helpful in prompting us to undertake a range of new investigations to support our conclusions and provide more data. We have also revised the text in places to avoid over-claiming on our findings and/or to add extra explanation (e.g. associating clinical phenotype to tissue-specific mutations).

Specifically:

1. In the first section focusing on promoters, the authors conclude that there is little evidence for promoter bivalency during human organogenesis. This is primarily based on their clustering analysis of promoter states. There are several concerns regarding this analysis:

We have undertaken further analyses which, as they have unfolded, also address the later points made by the Reviewer. In addition, we have toned down the manuscript text.

a. Clustering was performed using a too primarily oriented towards visualization and under default settings (more information should be provided on the precise parameters but in all available options the number of clusters seems to be determined by the user). Where is the evidence that the data can be optimally

described by five clusters? Would it be possible that allowing for more clusters would reveal bivalent promoters as a subset of the repressed promoters (especially given the faint H3K4me3 signal observed in the repressed state)?

Thank you for drawing this out. We would like to explain our logic and also describe the additional analyses that we have now undertaken.

We were indeed keen on effective visualisation that integrated the different datasets, i.e. correlated the co-occurrence of different histone modifications with gene expression level (Fig. 2a). Ngsplot allowed this and also generated relative quantitation of signal around the TSS rather than simple binary ‘yes/no’ calling of a particular histone mark (which might artificially imply equivalence for two marks when in fact one mark is far more prominent than the other and/or where small amounts of sample heterogeneity cannot be excluded). Very clearly, applying ngsplot default settings reliably depicted the major promoter states across tissues during human organogenesis. We too noted that H3K4me3 was on average slightly higher in the Active Repression/heavily H3K27me3 marked cluster. The choice of 5 clusters for the heatmaps was not user-defined but reflects default settings of the programme when applied to the three different histone modifications and the RNAseq. We have now explained these points in the manuscript text.

We appreciate the Reviewer’s request to explore additional subsets before drawing conclusions on bivalency (which we have now tempered in the manuscript). We have now sub-categorised the H3K27me3 signal to the limits of resolution prior to picking up common gene sets / cell types across tissues (after which drawing further conclusions would be wrong). Stipulating seven (kidney), eleven (liver) or ten (all other replicated tissues) categories in ngsplot allowed splitting of the original H3K27me3 grouping to include a small subset of genes with low level H3K4me3 signal—the putative bivalent signature that the Reviewer refers to. However, when we extracted the underlying genes and undertook functional analysis we did not see enrichment for imprinted genes or developmental TFs characteristic of a specific tissue fate. Instead, we observed the same annotations in all tissues for generic features like extracellular matrix and nerves. These data argue strongly towards mesenchymal/vascular and neural cells that are common across tissues rather than bivalency. Therefore, we still feel we have no evidence for bivalency on taking the analyses up to their limits of resolution. Benefiting from the Reviewer’s suggestions, we have tempered the original text and, in particular, highlighted these limits in our analysis. We have added more description and six new supplementary figures.

b. This concern becomes even more relevant if one takes a closer look into Figure 3a. Actively repressed promoters clearly show a bimodal behaviour. Some remain repressed across all assayed tissues whereas others switch between the expressed and repressed state. Could this group correspond to bivalent promoters?

We very much agree that some genes switch between ‘active repression’ and ‘expressed’ categories across tissues—this is the point we are trying to make with Fig. 3a and the analysis in Fig. 3b; these genes which show relief of repression from other sites (‘disallowed’) are enriched for key regulators of organ development (Fig. 3b). To move away over over-reliance on purely visualisation and to improve data-sharing, we have now tabulated these genes as Supplementary table 5. Theoretically, while these genes could be bivalently marked, we feel two points argue against it: firstly, they are in no way ‘poised’ in other tissues where they need to be excluded; secondly (more conclusively), they would have emerged in the new H3K27me3/H3K4me3 subcategory described above, and they have not.

Actually, we think the most likely reason that some key genes remain firmly H3K27me3 repressed across all our tissues (rather than switch to expressed) is because we simply haven’t yet sampled all organs in the human embryo at all developmental stages—i.e. these genes will be active somewhere else and / or at some other developmental time point. We made this point later in the manuscript when discussing the 9% overlap between H3K27me3 and the non-coding regions sequenced in the DDD study (lower part of page 6), but we have now made it more clearly earlier in the manuscript.

An easy way to check would be to split the two groups (globally repressed vs facultative repressed and compare the H3K4me3 signal between the two groups).

We feel that the new data above already address this point.

[Incidentally, we did wonder for a while whether H3K27me3 signal was elevated over the gene body of developmentally important TFs compared to other genes. However, this transpired to be largely artefactual because of their shorter average transcript length—the effect was greatly reduced / abolished by re-clustering around the TSS.]

c. Furthermore, since bivalent promoters are primarily described based on the co-occurrence of H3K4me3 and H3K27me3, it might be informative to additionally perform the clustering only considering these two modifications (and allowing for four or more clusters). H3K27ac and active transcription are not expected to be found in bivalent promoters so they are not particularly informative for this particular question but their presence might distort the clustering analysis.

While we understand this logic, now that we have isolated subgroups of genes with H3K27me3 and H3K4me3 (as above), we have already addressed this point. More broadly, we agree that H3K27ac would not be expected at bivalently marked promoters. Interestingly, the persistence of some H3K27ac in the newly identified H3K27me3/H3K4me3 subset (new Supplementary figures 6-11) is further support that we have not found bivalency, but that we have reached the limits of resolution for analysing our integrated data on promoter state.

2. In the last part of the manuscript, the authors intersect their putative enhancer annotation with a set of non-coding regions previously identified to be involved in developmental disorders. They then show that ~50% of noncoding de novo mutations fall within an active chromatin region. This is not surprising as these regions were previously selected for being conserved or for showing evidence of enhancer activity (experimentally validated or putative enhancers).

We agree this is not necessarily surprising but it is very reassuring. The experimentally validated enhancers pertained to only one tissue (heart), others were putative and conservation is not restricted to development (regulation of genes involved in terminally differentiated cell function is also conserved). Therefore, while we agree with the Reviewer, our analyses do reinforce that our data from human organogenesis ought to be very helpful when shared fully with the community.

A key step towards associating these mutations with the respective developmental phenotype would be to show that they are active in the organ that is most relevant to the respective disorder (i.e. neuronal organ for neurodevelopmental disorders and heart for heart defects). Surprisingly, with the exception of the single anecdotal case presented in Figure 7a, the authors don't attempt to associate organ-specific regulatory activity with the observed phenotype for each DNМ. Even in Figure 6d it seems that they test for enrichment of their entire H3K27ac set against different phenotypes (although I cannot be sure as this analysis is not properly documented in the methods).

We agree. This is a longer term ambition. We tried organ-level association with clinical phenotype. Within our team Matt Hurles was correct from the outset. Namely, that the DDD dataset, one of the largest clinical cohorts of its type, is insufficiently powered with patient numbers at present to allow this level of correlation, i.e. the challenge lies not with our human embryonic datasets but with the current scale of even the largest clinical cohorts. As described in the text, some correlation was observed for cases with neurodevelopmental delay (as this covered most of the DDD cohort) with the entire H3K27ac dataset. It is also important to note that these types of clinical cohorts from specific studies are very likely to bring some ascertainment bias. Moreover, minor neurodevelopmental issues present readily (to parents, schools) while minor anomalies elsewhere e.g. cardiac, lung, or renal often remain cryptic (e.g. bicuspid aortic valve). We have now re-written the text to make our points more clearly and to emphasise the future utility.

3. The authors should provide more information regarding their ChIP-seq to RNA-seq correlation analysis. This is briefly mentioned in the results section and is almost entirely absent from the methods. Which correlation metric was used? Did the authors correct for the effect of spurious correlations due to multiple testing (for example by shuffling regulatory regions and genes from distant chromosomes)? How was the significance of the observed correlations estimated? Shouldn't the results of this analysis be included in the Supplementary Material? This analysis cites Figure 7, which only presents data for one gene!

We have added more methods including our use of Pearson correlation.

Our major aim still stands. We want to showcase how useful we think our datasets will be in the future to clinical geneticists and others struggling to ascribe non-coding pathogenicity and / or prioritise target genes. We have not tried to claim significance for the correlations, but nonetheless feel that they provide a useful starting point to view and rank putative promoter and enhancer relationships with transcripts (only possible because of studying and integrating multiple tissues). We were only trying to exemplify this in Fig. 7—hence the single locus. However, we are making the complete genome-scale correlation data publicly available as additional UCSC tracks. The latter ought to be useful.

In a similar spirit, and while we wait for sufficient clinical noncoding sequence and phenotyping to permit fully powered association studies, e.g. in heart, we have now included proof-of-principle for inactivating a cardiac-specific enhancer which overlapped with a DDD region associated with congenital heart disease (new Figure 8). Although not explicitly requested, we undertook this by CRISPR-Cas9 in human pluripotent stem cells followed by differentiation to cardiomyocyte-like cells incorporating our existing NKX2-5 reporter system (referenced in the manuscript including Birket et al, Nat Biotech 2015). As can be seen in Fig. 8, cardiac differentiation was blocked. The enhancer correlated with expression of a distant gene, RBM24. Removal of the enhancer marked abrogated RBM24 expression. Other alterations were not significant (although CAP2 and ATXN1 appeared reduced). Notably, RBM24 expression was enriched in the heart compared to other datasets (60.5-fold). The gene has already been identified as a requirement for normal heart development—the knockout phenotype having multiple cardiac abnormalities including VSD (Yang et al, Dev Cell 2014, reference included). We are obviously not claiming this to be a systematic analysis of enhancer function but we hope that including these new data helps the reader to appreciate how our datasets may be useful to a wide community.

Minor comments:

1. It's quite alarming that some high rank patterns from the binary activity calling arise from combinations between single replicates of different organs (i.e. Fig. 5b upper limb/pancreas ranks 6th, adrenal and heart is 8th). Do the authors attribute this to the biological samples or to the approach used for calling activity? Could it also be related to differences in the developmental stages of the samples (i.e. clustering of samples from different tissues but more similar developmental stages)?

We disagree here. We do not feel our rankings of H3K27ac patterns across tissues are alarming but actually that they are likely to be important and useful. Hence, we purposefully chose not to hide high ranking single replicate associations (we could have easily filtered them out). Most importantly, tissue-specificity emerged in the top 0.6% of nearly 50,000 total patterns—to emphasise this we have now added greater detail than was in the original manuscript including a new Supplementary table 7. This percentage ranking points to very reproducible data and argues against any major contribution from variation in developmental stage or factors other than genuine biology. In fact, the combinations across tissues are interesting. Few analyses have looked in this way across organ systems at enhancer usage. While we specifically chose not to bias the data by only accepting replicated findings, users of the data could do this if they wished to re-run our analyses. The Reviewer mentions adrenal and heart. It is misleading to pick out single replicates ranking 8th when virtually all combinations involving replicates of the two organs ranked in the top 30 (8th, 13th, 21st, 25th and 27th i.e. all in the top 0.05% of all patterns). Please note that these are 1kb bins and that essentially the same mark will inevitably extend for somewhat variable distances across replicates. Further reassurance that we are capturing important biology comes from the tissue-specific patterns of TF binding motifs in Fig. 5c. These are heavily skewed towards TFs with major roles in the appropriate organ's development such as NR5A1 / SF-1 in the adrenal gland. Across tissues we see enrichment of GATA motifs for heart and pancreas (mutations in GATA4 and GATA6 cause congenital heart disease and pancreas agenesis). Taken together, we feel strongly that the data are robust and valuable.

2. Regarding the transgenic zebrafish experiment: It's not clear how sequence conservation in zebrafish was tested. Was this based on sequence only or was synteny also considered? The authors should state this in the methods. Scoring sequence conservation of non-coding elements over so long evolutionary distances is a difficult task and therefore absence of evidence for conservation should not be interpreted as evidence for its absence. Furthermore there is some ambiguity in the terms used by the authors (i.e. novel for previously unannotated, which could be confused with evolutionary novel; distance for evolutionary

distance, which in the context of enhancer function could be confused with physical distance). I would therefore recommend rephrasing the concluding statements as something similar to this: “These data imply that our H3K27ac detection marks previously unannotated human enhancers, which function over considerable evolutionary distance despite the absence of detectable sequence conservation. This is likely associated with the overall conserved TF activity between homologous organs (Stergachis et al. Nature 2014)”.

Thank you—this is a good point. We have clarified that we judged conservation by sequence. We agree that absence of evidence for conservation does not mean absence of conservation. We have adopted the suggested text in the revised manuscript. Thank you.

3. GO enrichments in temporal analysis: Not clear what the background is but it seems that all 19,700 genes in Figure 3a were used. Since the point of the analysis is to identify differences between genes derepressed in the transition from pluripotency to organogenesis vs from organogenesis to adult organ, it's more appropriate to limit the background set to genes expressed in the pancreas during any of the three stages. This might improve the resolution and remove confounders associated with general expression patterns in the pancreas (ex. involvement in diabetes).

We have clarified this in the Methods and figure legends (the original Methods text only really covered the analysis between embryonic tissues). The background is the full gene list. Our logic remains that this is the best approach because (i) this is what covers the ‘expressed’, ‘actively repressed’ and ‘inactive’ categories that comprise the alluvial plot and (ii) it is more conservative than picking out only pancreas-expressed genes. We feel that the latter point is quite important and makes the outcome (seeing independently enriched sets of early and late de-repressed genes) more reliable. As can be seen, this led to very informative gene ontology and we did not observe any evidence for confounders.

4. Overall, more information should be added to the methods section. The most critical issues were already highlighted above, but the authors could include more information in many sections (ex. parameters for MACS, STAR, read counting method, enrichment tests etc).

Thank you. We have tried to do this across the board.

In summary, Reviewer 2 has made some wide-ranging comments in amongst complimenting us on our work. We have tried our best to address them genuinely and openly above, undertaken new analyses and improved the manuscript. We hope the Reviewer is now content. We return to his/her initial comments that the dataset is impressive and valuable to a wide community. Accordingly, we are keen to make sure access to the data is as timely as possible.

Reviewer #3 (Remarks to the Author):

In this manuscript, Gerrard and colleagues mapped landscapes of histone H3K4me3, H3K27ac and H3K27me3 in thirteen tissues during human organogenesis. They classified discrete promoter states and identified a number of putative human embryo-enriched intergenic enhancers, several of which were validated in zebrafish. They further compared the promoter states and H3K27ac patterns across tissues. In addition, their analysis linked histone modification marked regions to de novo mutation sites which are associated with human developmental disorders.

Characterizing the epigenomic patterns during organogenesis is important for understanding the mechanisms of human organ development in health and disease. However, the author's conclusions on the epigenome analysis of the entire organ are not convincing, and such analysis did not provide useful information and insights into our understanding of human organ development. For example, the fetal liver contains > 20 cell types, including endoderm-derived cells and mesoderm-derived cells. These mixed ChIP-seq signals are not suitable for identifying cell type-specific DNA regulatory sequences. The human epigenetic atlas should be established at the cell lineage level or at the single cell level, but not at the organ level.

We agree with the value of single cell data and are part of the international Human Cell Atlas community that is starting to construct such analyses. Based on this knowledge, we disagree firmly with the ideology

that this negates the utility of data obtained in bulk. Very simply, the resolution we have acquired here is not possible in single cells at the moment. As single cell technology advances, a key reference point will be bulk datasets such as ours. Within our group we are applying this now for liver in analyses of scRNAseq (where technology remains compromised by drop out) and scATACseq (which lacks the equivalent epigenomic insight and depth of bulk analysis). It is also striking that 10X Genomics has recently informed the community of misanalysis that has arisen (and been published) using its version 1 software compromised by doublets.

We agree with the Editor's guidance on how to focus our rebuttal and revised manuscript.

Specifically, please ensure that the following requirements are met, and any relevant checklists are completed or updated and uploaded as a Related Manuscript file type with the revised article:

We have done this.

Finally, in response to the general journal comments on availability, we have made the raw data publicly available and deposited any new R codes on Github as cited in the manuscript. We have altered the previous layout containing this information to create a DATA AVAILABILITY section as requested. One of the most important outputs for us will be the full accessibility of our data so that others can re-use it. Based on the BioRxiv submission we have already started to share data with some groups.

In summary, thank you for all the comments above.

Reviewers' Comments:

Reviewer #2:

Remarks to the Author:

In this revised version of their manuscript, the authors:

1. Add more in-depth analyses to support their claim regarding the absence of promoter bivalency in cell-fate determining genes in human organogenesis.
2. Include sufficient details on how computational analyses were performed.
3. Provide the data for their correlation analysis between enhancer activity and gene expression, which will undoubtedly be a valuable resource to the research community.
4. Have appropriately toned down their claims and discuss limitations of their analysis (e.g., regarding the analysis of DDDs or the confounding effect of cellular heterogeneity).

Additionally, they generated more experimental data to support the functional relevance of the tissue-specific enhancers identified in this study.

The authors have thus sufficiently addressed all my concerns and I recommend this revised manuscript for publication.